# MODEL-BASED RL AS A MINIMALIST APPROACH TO HORIZON-FREE AND SECOND-ORDER BOUNDS

**Zhiyong Wang[1], Dongruo Zhou[2], John C.S. Lui[1], Wen Sun[3]**
[1]The Chinese University of Hong Kong
[2]Indiana University Bloomington
[3]Cornell University
`zywang21@cse.cuhk.edu.hk, dz13@iu.edu,`
`cslui@cse.cuhk.edu.hk, ws455@cornell.edu`

## ABSTRACT

Learning a transition model via Maximum Likelihood Estimation (MLE) followed by planning inside the learned model is perhaps the most standard and simplest Model-based Reinforcement Learning (RL) framework. In this work, we show that such a simple Model-based RL scheme, when equipped with optimistic and pessimistic planning procedures, achieves strong regret and sample complexity bounds in online and offline RL settings. Particularly, we demonstrate that under the conditions where the trajectory-wise reward is normalized between zero and one and the transition is time-homogenous, it achieves nearly horizon-free and second-order bounds.

## 1 INTRODUCTION

The framework of model-based Reinforcement Learning (RL) often consists of two steps: fitting a transition model using data and then performing planning inside the learned model. Such a simple framework turns out to be powerful and has been used extensively in practice on applications such as robotics and control (e.g., (Aboaf et al., 1989; Deisenroth et al., 2011; Venkatraman et al., 2017; Williams et al., 2017; Chua et al., 2018; Kaiser et al., 2019; Yang et al., 2023)).

The simplicity of model-based RL also attracts researchers to analyze its performance in settings such as online RL (Sun et al., 2019) and offline RL (Uehara & Sun, 2021). Mania et al. (2019) showed that this simple scheme — fitting model via data followed by optimal planning inside the model, has a strong performance guarantee under the classic linear quadratic regulator (LQR) control problems. Liu et al. (2023) showed that this simple MBRL framework when equipped with optimism in the face of the uncertainty principle, can achieve strong sample complexity bounds for a wide range of online RL problems with rich function approximation for the models. For offline settings where the model can only be learned from a static offline dataset, Uehara & Sun (2021) showed that MBRL equipped with the pessimism principle can again achieve robust performance guarantees for a large family of MDPs. Ross & Bagnell (2012) showed that in the hybrid RL setting where one has access to both online and offline data, this simple MBRL framework again achieves favorable performance guarantees without any optimism/pessimism algorithm design.

In this work, we do not create new MBRL algorithms, instead, we show that the extremely simple and standard MBRL algorithm – fitting models using Maximum Likelihood Estimation (MLE), followed by optimistic/pessimistic planning (depending on whether operating in online RL or offline RL mode), can already achieve surprising theoretical guarantees. Particularly, we show that under the conditions that trajectory-wise reward is normalized between zero and one, and the transition is time-homogenous, they can achieve nearly horizon-free and instance-dependent regret and sample complexity bounds, in both online and offline RL with non-linear function approximation. Nearly horizon-free bounds mean that the regret or sample complexity bounds have no explicit polynomial dependence on the horizon $H$. The motivation for studying horizon-free RL is to see if RL problems are harder than bandits due to the longer horizon planning in RL. *Our result here indicates that, even under non-linear function approximation, long-horizon planning is not the bottleneck of achieving statistical efficiency in RL.* For instance-dependent bounds, we focus on second-order bounds. A second-order regret bound scales with respect to the variances of the returns of policies and also directly implies a

first-order regret bound which scales with the expected reward of the optimal policy. Thus our instance-dependent bounds can be small under situations such as nearly-deterministic systems or the optimal policy having a small value. When specializing to the case of deterministic ground truth transitions (but the algorithm does not need to know this a priori), we show that these simple MBRL algorithms demonstrate a faster convergence rate than the worst-case rates. The key message of our work is

*Simple and standard MLE-based MBRL algorithms are sufficient for achieving nearly horizon-free and second-order bounds in online and offline RL with function approximation.*

We provide a fairly standard analysis to support the above claim. Our analysis follows the standard frameworks of optimism/pessimism in the face of uncertainty. For online RL. we use $\ell_1$ Eluder dimension (Liu et al., 2022; Wang et al., 2024a), a condition that uses both the MDP structure and the function class, to capture the structural complexity of exploration. For offline RL, we use the similar concentrability coefficient in Ye et al. (2024) to capture the coverage condition of the offline data. The key technique we leverage is the *triangular discrimination* – a divergence that is equivalent to the squared Hellinger distance up to some universal constants. Triangular discrimination was used in contextual bandit and model-free RL for achieving first-order and second-order instance-dependent bounds (Foster & Krishnamurthy, 2021; Wang et al., 2023, 2024a). Here we show that it also plays an important role in achieving horizon-free bounds. Our contributions can be summarized as follows.

1. Our results extend the scope of the prior work on horizon-free RL which only applies to tabular MDPs or MDPs with linear functions. Given a finite model class $\mathcal{P}$ (which could be exponentially large), we show that in online RL, the agent achieves an $O\left(\sqrt{(\sum_k \mathrm{VaR}_{\pi^k}) \cdot d_{\mathrm{RL}} \log(KH|\mathcal{P}|/\delta)} + d_{\mathrm{RL}} \log(KH|\mathcal{P}|/\delta)\right)$ regret, where $K$ is the number of episodes, $d_{\mathrm{RL}}$ is the $\ell_1$ Eluder dimension, $\mathrm{VaR}_{\pi^k}$ is the variance of the total reward of policy $\pi^k$ learned in episode $k$ and $\delta \in (0, 1)$ denotes the failure probability. Similarly, for offline RL, the agent achieves an $O\left(\sqrt{C^{\pi^*} \mathrm{VaR}_{\pi^*} \log(|\mathcal{P}|/\delta)/K} + C^{\pi^*} \log(|\mathcal{P}|/\delta)/K\right)$ performance gap in finding a comparator policy $\pi^*$, where $C^{\pi^*}$ is the single policy concentrability coefficient over $\pi^*$, $K$ denotes the number of offline trajectories, $\mathrm{VaR}_{\pi^*}$ is the variance of the total reward of $\pi^*$. For offline RL with finite $\mathcal{P}$, our result is *completely horizon-free*, not even with $\log H$ dependence.

2. When specializing to MDPs with deterministic ground truth transition (but rewards, and models in the model class could still be stochastic), we show that the same simple MBRL algorithms can adapt to the deterministic environment and achieve a better statistical complexity. For online RL, the regret becomes $O(d_{\mathrm{RL}} \log(KH|\mathcal{P}|/\delta))$, which only depends on the number of episodes $K$ poly-logarithmically. For offline RL, the performance gap to a comparator policy $\pi^*$ becomes $O\left(C^{\pi^*} \log(|\mathcal{P}|/\delta)/K\right)$, which is tighter than the worst-case $O(1/\sqrt{K})$ rate. All our results can be extended to continuous model class $\mathcal{P}$ using bracket number as the complexity measure.

Overall, we identify the *minimalist* algorithms and analysis for nearly horizon-free and instance-dependent (first & second-order) online & offline RL. By saying "*minimalist*" we mean the algorithm designs and analysis are much simpler than previous work on horizon-free and second-order RL.

## 2 RELATED WORK

**Model-based RL.** Learning transition models with function approximation and planning with the learned model is a standard approach in RL and control. In the control literature, certainty-equivalence control learns a model from some data and plans using the learned model, which is simple but effective for controlling systems such as Linear Quadratic Regulators (LQRs) (Mania et al., 2019). In RL, such a simple model-based framework has been widely used in theory with rich function approximation, for online RL (Sun et al., 2019; Foster et al., 2021; Song & Sun, 2021; Zhan et al., 2022; Liu et al., 2022, 2023; Zhong et al., 2022), offline RL (Uehara & Sun, 2021), RL with representation learning (Agarwal et al., 2020; Uehara et al., 2021), and hybrid RL using both online and offline data for model fitting (Ross & Bagnell, 2012). Our work builds on the maximum-likelihood estimation (MLE) approach, a standard method for estimating transition models in model-based RL.

**Horizon-free and Instance-dependent bounds.** Most existing works on horizon-free RL typically focus on tabular settings or linear settings. For instance, Wang et al. (2020) firstly studied horizon-free RL for tabular MDPs and proposed an algorithm that depends on horizon logarithmically. Several follow-up work studied horizon-free RL for tabular MDP with better sample complexity (Zhang et al., 2021a), offline RL (Ren et al., 2021), stochastic shortest path (Tarbouriech et al., 2021) and

RL with linear function approximation (Kim et al., 2022; Zhang et al., 2021b; Zhou & Gu, 2022; Di et al., 2023; Zhang et al., 2024b, 2023; Zhao et al., 2023b). Note that all these works have logarithmic dependence on the horizon $H$. For the tabular setting, recent work further improved the regret or sample complexity to be completely independent of the horizon (i.e., removing the logarithmic dependence on the horizon) (Li et al., 2022; Zhang et al., 2022) with a worse dependence on the cardinality of state and action spaces $|\mathcal{S}|$ and $|\mathcal{A}|$. A recent work Li & Yang (2023) further improved the dependence on $|\mathcal{S}|$ and $|\mathcal{A}|$. To compare with, we show that simple MBRL algorithms are already enough to achieve completely horizon-free (i.e., no log dependence) sample complexity for offline RL when the transition model class is finite, and we provide a simpler approach to achieve the nearly horizon-free results for tabular MDPs, compared with Zhang et al. (2021a). A recent work (Huang et al., 2024) also studied the horizon-free and instance-dependent online RL in the function approximation setting with small Eluder dimensions. They estimated the variances to conduct variance-weighted regression. To compare, in our online RL part, we use the simple and standard MLE-based MBRL approach and analysis to get similar guarantees. A more recent work also studied horizon-free behavior cloning Foster et al. (2024), which is different from our settings.

Besides horizon-free RL, another line of work aimed to provide algorithms with instance-dependent sample complexity/regret bounds, which often enjoy tighter statistical complexity compared with previous work. To mention a few, Zanette & Brunskill (2019) proposed an EULER algorithm with an instance-dependent regret which depends on the maximum variance of the policy return over all policies. Later, Wagenmaker et al. (2022); Wang et al. (2023) proposed algorithms with first-order regret bounds. A more refined second-order regret bound has been studied. The second-order regret bound is a well-studied instance-dependent bound in the online learning and bandit literature (Cesa-Bianchi et al., 2007; Ito et al., 2020; Olkhovskaya et al., 2024; Wang et al., 2024b), and compared to the bounds that depend on the maximum variance over all policies, it can be much smaller and it also implies a first-order regret bound. Zhang et al. (2024a); Zhou et al. (2023) proposed algorithms for tabular MDP with second-order regret bounds. Zhao et al. (2023b) studied the RL with linear function approximation and proposed algorithms with both horizon-free and variance-dependent regret bounds. The closest work to us is Wang et al. (2023, 2024a), which used model-free distributional RL methods to achieve first-order and second-order regret bounds in RL. Their approach relies on distributional RL, a somewhat non-conventional approach for RL. Their regret bounds have explicit polynomial dependence on the horizon. Our work focuses on the more conventional model-based RL algorithms and demonstrates that they are indeed sufficient to achieve horizon-free and second-order regret bounds.

## 3 PRELIMINARIES

**Markov Decision Processes.** We consider finite horizon time-homogenous MDP $\mathcal{M} = \{\mathcal{S}, \mathcal{A}, H, P^\star, r, s_0\}$ where $\mathcal{S}, \mathcal{A}$ are the state and action space (could be large or even continuous), $H \in \mathbb{N}^+$ is the horizon, $P^\star : \mathcal{S} \times \mathcal{A} \mapsto \Delta(\mathcal{S})$ is the ground truth transition, $r : \mathcal{S} \times \mathcal{A} \mapsto \mathbb{R}$ is the reward signal which we assume is known to the learner, and $s_0$ is the fixed initial state.[1] Note that the transition $P^\star$ here is time-homogenous. For notational easiness, we denote $[K-1] = \{0, 1, \ldots, K-1\}$.

We denote $\pi$ as a deterministic non-stationary policy $\pi = \{\pi_0, \ldots, \pi_{H-1}\}$ where $\pi_h : \mathcal{S} \mapsto \mathcal{A}$ maps from a state to an action. Let $\Pi$ denote the set of all such policies. $V_h^\pi(s)$ represents the expected total reward of policy $\pi$ starting at $s_h = s$, and $Q_h^\pi(s, a)$ is the expected total reward of the process of executing $a$ at $s$ at time step $h$ followed by executing $\pi$ to the end. The optimal policy $\pi^\star$ is defined as $\pi^\star = \arg\max_\pi V_0^\pi(s_0)$. For notation simplicity, we denote $V^\pi := V_0^\pi(s_0)$. We will denote $d_h^\pi(s, a)$ as the state-action distribution induced by policy $\pi$ at time step $h$. We sometimes will overload notation and denote $d_h^\pi(s)$ as the corresponding state distribution at $h$. Sampling $s \sim d_h^\pi$ means executing $\pi$ starting from $s_0$ to $h$ and returning the state at time step $h$.

Since we use the model-based approach for learning, we define a general model class $\mathcal{P} \subset \mathcal{S} \times \mathcal{A} \mapsto \Delta(\mathcal{S})$. Given a transition $P$, we denote $V_{h;P}^\pi$ and $Q_{h;P}^\pi$ as the value and Q functions of policy $\pi$ under the model $P$. Given a function $f : \mathcal{S} \times \mathcal{A} \mapsto \mathbb{R}$, we denote the $(Pf)(s, a) := \mathbb{E}_{s' \sim P(s,a)} f(s')$. We then denote the *variance induced by one-step transition $P$ and function $f$* as $(\mathbb{V}_P f)(s, a) := (Pf^2)(s, a) - (Pf(s, a))^2$ which is equal to $\mathbb{E}_{s' \sim P(s,a)} f^2(s') - (\mathbb{E}_{s' \sim P(s,a)} f(s'))^2$.

---

[1] For simplicity, we assume initial state $s_0$ is fixed and known. Our analysis can be easily extended to a setting where the initial state is sampled from an unknown fixed distribution.

**Assumptions.** We make the realizability assumption that $P^\star \in \mathcal{P}$. We assume that the rewards are normalized such that $r(\tau) \in [0, 1]$ for any trajectory $\tau := \{s_0, a_0, \ldots, s_{H-1}, a_{H-1}\}$ where $r(\tau)$ is short for $\sum_{h=0}^{H-1} r(s_h, a_h)$. Note that this setting is more general than assuming each one-step reward is bounded, i.e., $r(s_h, a_h) \in [0, 1/H]$, and allows to represent the sparse reward setting. Without loss of generalizability, we assume $V_{h;P}^\pi(s) \in [0, 1]$, for all $\pi \in \Pi, h \in [0, H], P \in \mathcal{P}, s \in \mathcal{S}^2$.

**Online RL.** For the online RL setting, we focus on the episodic setting where the learner can interact with the environment for $K$ episodes. At episode $k$, the learner proposes a policy $\pi^k$ (based on the past interaction history), executes $\pi^k$ starting from $s_0$ to time step $H - 1$. We measure the performance of the online learning via *regret*: $\sum_{k=0}^{K-1} \left( V^{\pi^\star} - V^{\pi^k} \right)$. To achieve meaningful regret bounds, we often need additional structural assumptions on the MDP and the model class $\mathcal{P}$. We use a $\ell_1$ Eluder dimension (Liu et al., 2022) as the structural condition due to its ability to capture non-linear function approximators (formal definition will be given in Section 4).

**Offline RL.** For the offline RL setting, we assume that we have a pre-collected offline dataset $\mathcal{D} = \{\tau^i\}_{i=1}^K$ which contains $K$ trajectories [3]. For each trajectory, we allow it to potentially be generated by an adversary, i.e., at step $h$ in trajectory $k$, (i.e., $s_h^k$), the adversary can select $a_h^k$ based on all history (the past $k - 1$ trajectories and the steps before $h$ within trajectory $k$) with a fixed strategy, with the only condition that the state transitions follow the underlying transition dynamics, i.e., $s_{h+1}^i \sim P^\star(s_h^i, a_h^i)$. We emphasize that $\mathcal{D}$ is not necessarily generated by some offline trajectory distribution. Given $\mathcal{D}$, we can split the data into $HK$ many state-action-next state $(s, a, s')$ tuples which we can use to learn the transition. To succeed in offline learning, we typically require the offline dataset to have good coverage over some high-quality comparator policy $\pi^*$ (formal definition of coverage will be given in Section 5). Our goal here is to learn a policy $\hat{\pi}$ that is as good as $\pi^*$, and we are interested in the *performance gap* between $\hat{\pi}$ and $\pi^*$, i.e., $V^{\pi^*} - V^{\hat{\pi}}$.

**Horizon-free and Second-order Bounds.** Our goal is to achieve regret bounds (online RL) or performance gaps (offline RL) that are (nearly) horizon-free, i.e., logarithmic dependence on $H$. In addition to the horizon-free guarantee, we also want our bounds to scale with respect to the variance of the policies. Denote $\mathrm{VaR}_\pi$ as the variance of trajectory reward, i.e., $\mathrm{VaR}_\pi := \mathbb{E}_{\tau \sim \pi}(r(\tau) - \mathbb{E}_{\tau \sim \pi} r(\tau))^2$. Second-order bounds in offline RL scales with $\mathrm{VaR}_{\pi^*}$ – the variance of the comparator policy. Second-order regret bound in online setting scales with respect to $\sqrt{\sum_k \mathrm{VaR}_{\pi^k}}$ instead of $\sqrt{K}$. Note that in the worst case, $\sqrt{\sum_k \mathrm{VaR}_{\pi^k}}$ scales in the order of $\sqrt{K}$, but can be much smaller in benign cases such as nearly deterministic MDPs. We also note that second-order regret bound immediately implies first-order regret bound in the reward maximization setting, which scales in the order $\sqrt{KV^{\pi^\star}}$ instead of just $\sqrt{K}$. The first order regret bound $\sqrt{KV^{\pi^\star}}$ is never worse than $\sqrt{K}$ since $V^{\pi^\star} \leq 1$. Thus, by achieving a second-order regret bound, our algorithm immediately achieves a first-order regret bound.

**Additional notations.** Given two distributions $p \in \Delta(\mathcal{X})$ and $q \in \Delta(\mathcal{X})$, we denote the triangle discrimination $D_\triangle(p \parallel q) = \sum_{x \in \mathcal{X}} \frac{(p(x)-q(x))^2}{p(x)+q(x)}$, and squared Hellinger distance $\mathbb{H}^2(p \parallel q) = \frac{1}{2} \sum_{x \in \mathcal{X}} \left( \sqrt{q(x)} - \sqrt{p(x)} \right)^2$ (we replace sum via integral when $\mathcal{X}$ is continuous and $p$ and $q$ are pdfs). Note that $D_\triangle$ and $\mathbb{H}^2$ are equivalent up to universal constants. We will frequently use the following key lemma in (Wang et al., 2024a) to control the difference between means of two distributions.

**Lemma 1** (Lemma 4.3 in Wang et al. (2024a)). *For two distributions $f \in \Delta([0, 1])$ and $g \in \Delta([0, 1])$:*

$$|\mathbb{E}_{x \sim f}[x] - \mathbb{E}_{x \sim g}[x]| \leq 4\sqrt{\mathrm{VaR}_f \cdot D_\triangle(f \parallel g)} + 5D_\triangle(f \parallel g). \tag{1}$$

*where $\mathrm{VaR}_f := \mathbb{E}_{x \sim f}(x - \mathbb{E}_{x \sim f}[x])^2$ denotes the variance of the distribution $f$.*

The lemma plays a key role in achieving second-order bounds (Wang et al., 2024a). The intuition is the means of the two distributions can be closer if one of the distributions has a small variance. A more naive way of bounding the difference in means is $|\mathbb{E}_{x \sim f}[x] - \mathbb{E}_{x \sim g}[x]| \leq (\max_{x \in \mathcal{X}} |x|) \|f - g\|_1 \lesssim$

---

[2] $r(\tau) \in [0, 1]$ implies $V_{h;P^\star}^\pi(s) \in [0, 1]$. If we do not assume $V_{h;P}^\pi(s) \in [0, 1]$ for all $P \in \mathcal{P}$, we can simply add a filtering step in the algorithm to only choose $\pi, P$ with $V_{h;P}^\pi(s_0) \in [0, 1]$ to get the same guarantees.

[3] Our algorithms and analysis can also apply to the case where the dataset only consists of transitions, as they rely solely on transition-level information from the offline dataset

---

**Algorithm 1** Optimistic Model-based RL (O-MBRL)

---

1: **Input:** model class $\mathcal{P}$, confidence parameter $\delta \in (0, 1)$, threshold $\beta$.
2: Initialize $\pi^0$, initialize dataset $\mathcal{D} = \varnothing$.
3: **for** $k = 0 \to K - 1$ **do**
4:    Collect a trajectory $\tau = \{s_0, a_0, \cdots, s_{H-1}, a_{H-1}\}$ from $\pi^k$, split it into tuples of $\{s, a, s'\}$ and add to $\mathcal{D}$.
5:    Construct a version space $\widehat{\mathcal{P}}^k$:

$$\widehat{\mathcal{P}}^k = \left\{ P \in \mathcal{P} : \sum_{s,a,s' \in \mathcal{D}} \log P(s'_i | s_i, a_i) \geq \max_{\tilde{P} \in \mathcal{P}} \sum_{s,a,s' \in \mathcal{D}} \log \tilde{P}(s'_i | s_i, a_i) - \beta \right\}.$$

6:    Set $(\pi^k, \widehat{P}^k) \leftarrow \operatorname{argmax}_{\pi \in \Pi, P \in \widehat{\mathcal{P}}^k} V^\pi_{0;P}(s_0)$.
7: **end for**

---

$(\max_{x \in \mathcal{X}} |x|) \mathbb{H}(f \parallel g) \lesssim (\max_{x \in \mathcal{X}} |x|) \sqrt{D_\triangle(f \parallel g)}$. Such an approach would have to pay the maximum range $\max_{x \in \mathcal{X}} |x|$ and thus can not leverage the variance $\operatorname{VaR}_f$. In the next sections, we show this lemma plays an important role in achieving horizon-free and second-order bounds.

## 4 ONLINE SETTING

In this section, we study the online setting. We present the optimistic model-based RL algorithm (O-MBRL) in Algorithm 1. The algorithm starts from scratch, and iteratively maintains a version space $\widehat{\mathcal{P}}^k$ of the model class using the historical data collected so far. Again the version space is designed such that for all $k \in [0, K-1]$, we have $P^\star \in \widehat{\mathcal{P}}_k$ with high probability. The policy $\pi^k$ in this case is computed via the optimism principle, i.e., it selects $\pi^k$ and $\widehat{P}^k$ such that $V^{\pi^k}_{\widehat{\mathcal{P}}^k} \geq V^{\pi^\star}$.

Note that the algorithm design in Algorithm 1 is not new and in fact is quite standard in the model-based RL literature. For instance, Sun et al. (2019) presented a similar style of algorithm except that they use a min-max GAN style objective for learning models. Zhan et al. (2022) used MLE oracle with optimism planning for Partially observable systems such as Predictive State Representations (PSRs), and Liu et al. (2023) used them for both partially and fully observable systems. However, their analyses do not give horizon-free and instance-dependent bounds. We show that under the structural condition that captures nonlinear function class with small eluder dimensions, Algorithm 1 achieves horizon-free and second-order bounds. Besides, since second-order regret bound implies first-order bound (Wang et al., 2024a), our result immediately implies a first-order bound as well.

We first introduce the $\ell_p$ Eluder dimension as follows.

**Definition 1** ($\ell_p$ Eluder Dimension). *$DE_p(\Psi, \mathcal{X}, \epsilon)$ is the eluder dimension for $\mathcal{X}$ with function class $\Psi$, when the longest $\epsilon$-independent sequence $x^1, \ldots, x^L \subseteq \mathcal{X}$ enjoys the length less than $DE_p(\Psi, \mathcal{X}, \epsilon)$. We say that a sequence $x^1, \ldots, x^L \subseteq \mathcal{X}$ is $\epsilon$-independent if there exists $g \in \Psi$ such that for all $t \in [L]$, $\sum_{l=1}^{t-1} |g(x^l)|^p \leq \epsilon^p$ and $|g(x^t)| > \epsilon$.*

We work with the $\ell_1$ Eluder dimension $DE_1(\Psi, \mathcal{S} \times \mathcal{A}, \epsilon)$ with the function class $\Psi$ specified as:

$$\Psi = \left\{ (s, a) \mapsto \mathbb{H}^2(P^\star(s, a) \parallel P(s, a)) : P \in \mathcal{P} \right\}.$$

**Remark 1.** *The $\ell_1$ Eluder dimension has been used in previous works such as Liu et al. (2022). We have the following corollary to demonstrate that the $\ell_1$ dimension generalizes the original $\ell_2$ dimension of Russo & Van Roy (2013), it can capture tabular, linear, and generalized linear models.*

**Lemma 2** (Proposition 19 in (Liu et al., 2022)). *For any $\Psi, \mathcal{X}, \epsilon > 0$, $\mathrm{DE}_1(\Psi, \mathcal{X}, \epsilon) \leq \mathrm{DE}_2(\Psi, \mathcal{X}, \epsilon)$.*

We are ready to present our main theorem for the online RL setting with finite class $\mathcal{P}$.

**Theorem 1** (Main theorem for online setting with finite $\mathcal{P}$). *For any $\delta \in (0, 1)$, let $\beta = 4 \log \left( \frac{K|\mathcal{P}|}{\delta} \right)$, with probability at least $1 - \delta$, Algorithm 1 achieves the following regret bound:*

$$\sum_{k=0}^{K-1} (V^{\pi^\star} - V^{\pi^k}) \leq O\Big( \sqrt{\sum_{k=0}^{K-1} \operatorname{VaR}_{\pi^k} \cdot DE_1(\Psi, \mathcal{S} \times \mathcal{A}, 1/KH) \cdot \log(KH|\mathcal{P}|/\delta) \log(KH)}$$
$$+ DE_1(\Psi, \mathcal{S} \times \mathcal{A}, 1/KH) \cdot \log(KH|\mathcal{P}|/\delta) \log(KH) \Big). \tag{2}$$

The above theorem indicates the standard and simple O-MBRL algorithm is already enough to achieve horizon-free and second-order regret bounds: our bound does not have explicit polynomial dependences on horizon $H$, the leading term scales with $\sqrt{\sum_k \text{VaR}_{\pi^k}}$ instead of the typical $\sqrt{K}$.

We have the following result about the first-order regret bound.

**Corollary 1** (Horizon-free and First-order regret bound). *Let $\beta = 4\log\left(\frac{K|\mathcal{P}|}{\delta}\right)$, with probability at least $1 - \delta$, Algorithm 1 achieves the following regret bound:*

$$\sum_{k=0}^{K-1} V^{\pi^\star} - V^{\pi^k} \leq O\Big(\sqrt{KV^{\pi^\star} \cdot DE_1(\Psi, \mathcal{S} \times \mathcal{A}, 1/KH) \cdot \log(KH|\mathcal{P}|/\delta)\log(KH)}$$
$$+ DE_1(\Psi, \mathcal{S} \times \mathcal{A}, 1/KH) \cdot \log(KH|\mathcal{P}|/\delta)\log(KH)\Big).$$

*Proof.* Note that $\text{VaR}_\pi \leq V^\pi \leq V^{\pi^\star}$ where the first inequality is because the trajectory-wise reward is bounded in $[0, 1]$. Therefore, combining with Theorem 1, we directly obtain the first-order result. $\square$

Note that the above bound scales with respect to $\sqrt{KV^{\pi^\star}}$ instead of just $\sqrt{K}$. Since $V^{\pi^\star} \leq 1$, this bound improves the worst-case regret bound when the optimal policy has total reward less than one.[4]

**Faster rates for deterministic transitions.** When the underlying MDP has deterministic transitions, we can achieve a smaller regret bound that only depends on the number of episodes logarithmically.

**Corollary 2** ($\log K$ regret bound with deterministic transitions). *When the transition dynamics of the MDP are deterministic, setting $\beta = 4\log\left(\frac{K|\mathcal{P}|}{\delta}\right)$, w.p. at least $1 - \delta$, Algorithm 1 achieves:*

$$\sum_{k=0}^{K-1} V^{\pi^\star} - V^{\pi^k} \leq O\left(DE_1(\Psi, \mathcal{S} \times \mathcal{A}, 1/KH) \cdot \log(KH|\mathcal{P}|/\delta)\log(KH)\right).$$

**Extension to infinite class $\mathcal{P}$.** For infinite model class $\mathcal{P}$, we have a similar result. First, we define the bracketing number of an infinite model class as follows.

**Definition 2** (Bracketing Number (Geer, 2000)). *Let $\mathcal{G}$ be a set of functions mapping $\mathcal{X} \to \mathbb{R}$. Given two functions $l, u$ such that $l(x) \leq u(x)$ for all $x \in \mathcal{X}$, the bracket $[l, u]$ is the set of functions $g \in \mathcal{G}$ such that $l(x) \leq g(x) \leq u(x)$ for all $x \in \mathcal{X}$. We call $[l, u]$ an $\epsilon$-bracket if $\|u - l\| \leq \epsilon$. Then, the $\epsilon$-bracketing number of $\mathcal{G}$ with respect to $\|\cdot\|$, denoted by $\mathcal{N}_{[]}(\epsilon, \mathcal{G}, \|\cdot\|)$ is the minimum number of $\epsilon$-brackets needed to cover $\mathcal{G}$.*

As an example, according to Appendix C.2 of Uehara & Sun (2021), the bracketing number of Linear Mixture MDPs is $\mathcal{N}_{[]}(\epsilon, \mathcal{P}, \|\cdot\|) = O(1/\epsilon)^d$. We use the bracketing number of $\mathcal{P}$ to denote the complexity of the model class, similar to $|\mathcal{P}|$ in the finite class case. Next, we propose a corollary to characterize the regret with an infinite model class.

**Corollary 3** (Regret bound for Algorithm 1 with infinite model class $\mathcal{P}$). *When the model class $\mathcal{P}$ is infinite, let $\beta = 7\log(K\mathcal{N}_{[]}((KH|\mathcal{S}|)^{-1}, \mathcal{P}, \|\cdot\|_\infty)/\delta)$, with probability at least $1 - \delta$, Algorithm 1 achieves the following regret bound:*

$$\sum_{k=0}^{K-1} V^{\pi^\star} - V^{\pi^k} \leq O\Bigg(DE_1(\Psi, \mathcal{S} \times \mathcal{A}, \frac{1}{KH})\log(\frac{KH\mathcal{N}_{[]}((KH|\mathcal{S}|)^{-1}, \mathcal{P}, \|\cdot\|_\infty)}{\delta})\log(KH)$$
$$+ \sqrt{\sum_{k=0}^{K-1} \text{VaR}_{\pi^k} \cdot DE_1(\Psi, \mathcal{S} \times \mathcal{A}, \frac{1}{KH})\log(\frac{KH\mathcal{N}_{[]}((KH|\mathcal{S}|)^{-1}, \mathcal{P}, \|\cdot\|_\infty)}{\delta})\log(KH)}\Bigg),$$

*where $\mathcal{N}_{[]}((KH|\mathcal{S}|)^{-1}, \mathcal{P}, \|\cdot\|_\infty)$ is the bracketing number defined in Definition 2.*

A specific example of the infinite model class is the tabular MDP, where $\mathcal{P}$ is the collection of all the conditional distributions over $\mathcal{S} \times \mathcal{A} \to \Delta(\mathcal{S})$. By Corollary 3, we also have a new regret bound for MBRL under the tabular MDP setting, which is nearly horizon-free and second-order.

---

[4]Typically a first-order regret bound makes more sense in the cost minimization setting instead of reward maximization setting. We believe that our results are transferable to the cost-minimization setting.

**Example 1** (Tabular MDPs). *When specializing to tabular MDPs, use the fact that tabular MDP has $\ell_2$ Eluder dimension being at most $|\mathcal{S}||\mathcal{A}|$ (Section D.1 in Russo & Van Roy (2013)), $\ell_1$ dimension is upper bounded by $\ell_2$ dimension (Lemma 2), and use the standard $\epsilon$-net argument to show that $\mathcal{N}_{[]}(\epsilon, \mathcal{P}, \|\cdot\|_\infty)$ is upper-bounded by $(c/\epsilon)^{|\mathcal{S}|^2|\mathcal{A}|}$ (e.g., see Uehara & Sun (2021)), we can show that Algorithm 1 achieves the following regret bound for tabular MDP: with probability at least $1 - \delta$,*

$$\sum_k V^{\pi^\star} - V^{\pi^k} \le O\Big(|\mathcal{S}|^{1.5}|\mathcal{A}|\sqrt{\sum_k \mathrm{VaR}_{\pi^k} \cdot \log(\frac{KH|\mathcal{S}|}{\delta})}\log(KH) + |\mathcal{S}|^3|\mathcal{A}|^2\log(\frac{KH|\mathcal{S}|}{\delta})\log(KH)\Big).$$

In summary, we have shown that a simple MLE-based MBRL algorithm is enough to achieve nearly horizon-free and second-order regret bounds under non-linear function approximation.

### 4.1 PROOF SKETCH OF THEOREM 1

Now we are ready to provide a proof sketch of Theorem 1 with the full proof deferred to Appendix D.1. For ease of presentation, we use $d_{\mathsf{RL}}$ to denote $\mathrm{DE}_1(\Psi, \mathcal{S} \times \mathcal{A}, 1/KH)$, and ignore some log terms.

Overall, our analysis follows the general framework of optimism in the face of uncertainty, but with (1) careful analysis in leveraging the MLE generalization bound and (2) more refined proof in the training-to-testing distribution transfer via Eluder dimension.

By standard MLE analysis, we can show w.p. $1 - \delta$, for all $k \in [K-1]$, we have $P^\star \in \widehat{\mathcal{P}}^k$, and

$$\sum_{i=0}^{k-1} \sum_{h=0}^{H-1} \mathbb{H}^2(P^\star(s_h^i, a_h^i)\|\widehat{P}^k(s_h^i, a_h^i)) \le O(\log(K|\mathcal{P}|/\delta)). \tag{3}$$

From here, trivially applying training-to-testing distribution transfer via the Eluder dimension as previous works (e.g., Wang et al. (2024a)) would cause poly-dependence on $H$. With new techniques detailed in Appendix B, which is one of our technical contributions and may be of independent interest, we can get: there exists a set $\mathcal{K} \subseteq [K-1]$ such that $|\mathcal{K}| \le O(d_{\mathsf{RL}} \log(K|\mathcal{P}|/\delta))$, and

$$\sum_{k \in [K-1] \setminus \mathcal{K}} \sum_h \mathbb{H}^2\Big(P^\star(s_h^k, a_h^k) \,\|\, \widehat{P}^k(s_h^k, a_h^k)\Big) \le O(d_{\mathsf{RL}} \cdot \log(K|\mathcal{P}|/\delta)\log(KH)). \tag{4}$$

Recall that $(\pi^k, \widehat{P}^k) \leftarrow \mathrm{argmax}_{\pi \in \Pi, P \in \widehat{\mathcal{P}}^k} V_{0;P}^\pi(s_0)$, with the above realization guarantee $P^\star \in \widehat{\mathcal{P}}^k$, we can get the following optimism guarantee: $V_{0;P^\star}^\star \le \max_{\pi \in \Pi, P \in \widehat{\mathcal{P}}^k} V_{0;P}^\pi = V_{0;\widehat{P}^k}^{\pi^k}$.

At this stage, one straight-forward way to proceed is to use the standard simulation lemma (Lemma 7):

$$\sum_{k=0}^{K-1} V_{0;\widehat{P}^k}^{\pi^k} - V_{0;P^\star}^{\pi^k} \le \sum_{k=0}^{K-1} \sum_{h=0}^{H-1} \mathbb{E}_{s,a \sim d_h^{\pi^k}} \left[\left|\mathbb{E}_{s' \sim P^\star(s,a)} V_{h+1;\widehat{P}^k}^{\pi^k}(s') - \mathbb{E}_{s' \sim \widehat{P}^k(s,a)} V_{h+1;\widehat{P}^k}^{\pi^k}(s')\right|\right]. \tag{5}$$

However, from here, if we naively bound each term on the RHS via $\mathbb{E}_{s,a \sim d_h^{\pi^k}} \|P^\star(s,a) - \widehat{P}(s,a)\|_1$, which is what previous works such as Uehara & Sun (2021) did exactly, we would end up paying a linear horizon dependence $H$ due to the summation over $H$ on the RHS the above expression. Given the mean-to-variance lemma (Lemma 1), we may consider using it to bound the difference between two means $\mathbb{E}_{s' \sim P^\star(s,a)} V_{h+1;\widehat{P}^k}^{\pi^k}(s') - \mathbb{E}_{s' \sim \widehat{P}^k(s,a)} V_{h+1;\widehat{P}^k}^{\pi^k}(s')$. This still can not work if we start from here, because we would eventually get $\sum_k \sum_h \mathbb{E}_{s,a \sim d_h^{\pi^k}}[\mathbb{H}^2(P^\star(s,a)\|\widehat{P}^k(s,a))]$ terms, which can not be further upper bounded easily with the MLE generalization guarantee.

To achieve horizon-free and second-order bounds, we need a novel and more careful analysis.

First, we carefully decompose and upper bound the regret in $\tilde{\mathcal{K}} := [K-1] \setminus \mathcal{K}$ w.h.p. as follows using Bernstain's inequality (for regret in $\mathcal{K}$ we simply upper bound it by $|\mathcal{K}|$)

$$\sum_{k \in \tilde{\mathcal{K}}} \left(V_{0;\widehat{P}^k}^{\pi^k}(s_h^k) - \sum_{h=0}^{H-1} r(s_h^k, a_h^k)\right) + \sum_{k \in \tilde{\mathcal{K}}} \left(\sum_{h=0}^{H-1} r(s_h^k, a_h^k) - V_{0;P^\star}^{\pi^k}\right) \lesssim \sqrt{\sum_{k \in \tilde{\mathcal{K}}} \sum_h \left(\mathbb{V}_{P^\star} V_{h+1;\widehat{P}^k}^{\pi^k}\right)(s_h^k, a_h^k)}$$

$$+ \sum_{k \in \tilde{\mathcal{K}}} \sum_h \left|\mathbb{E}_{s' \sim \widehat{P}^k(s_h^k, a_h^k)} V_{h+1;\widehat{P}^k}^{\pi^k}(s') - \mathbb{E}_{s' \sim P^\star(s_h^k, a_h^k)} V_{h+1;\widehat{P}^k}^{\pi^k}(s')\right| + \sqrt{\sum_k \mathrm{VaR}_{\pi^k} \log(1/\delta)}. \tag{6}$$

Then, we bound the difference of two means $\mathbb{E}_{s' \sim \widehat{P}^k(s_h^k, a_h^k)} V_{h+1;\widehat{P}^k}^{\pi^k}(s') - \mathbb{E}_{s' \sim P^\star(s_h^k, a_h^k)} V_{h+1;\widehat{P}^k}^{\pi^k}(s')$ using variances and the triangle discrimination (see Lemma 1 for more details), together with the fact that $D_\triangle \leq 4\mathbb{H}^2$, and information processing inequality on the squared Hellinger distance, we have

$$
\left| \mathbb{E}_{s' \sim \widehat{P}^k(s_h^k, a_h^k)} V_{h+1;\widehat{P}^k}^{\pi^k}(s') - \mathbb{E}_{s' \sim P^\star(s_h^k, a_h^k)} V_{h+1;\widehat{P}^k}^{\pi^k}(s') \right|
$$

$$
\leq O\Big( \sqrt{ \left( \mathbb{V}_{P^\star} V_{h+1;\widehat{P}^k}^{\pi^k} \right)(s_h^k, a_h^k) D_\triangle \left( V_{h+1;\widehat{P}^k}^{\pi^k}(s' \sim P^\star(s_h^k, a_h^k)) \parallel V_{h+1;\widehat{P}^k}^{\pi^k}(s' \sim \widehat{P}^k(s_h^k, a_h^k)) \right) }
$$

$$
+ D_\triangle \left( V_{h+1;\widehat{P}^k}^{\pi^k}(s' \sim P^\star(s_h^k, a_h^k)) \parallel V_{h+1;\widehat{P}^k}^{\pi^k}(s' \sim \widehat{P}^k(s_h^k, a_h^k)) \right) \Big)
$$

$$
\leq O\Big( \sqrt{ \left( \mathbb{V}_{P^\star} V_{h+1;\widehat{P}^k}^{\pi^k} \right)(s_h^k, a_h^k) \mathbb{H}^2 \left( P^\star(s_h^k, a_h^k) \parallel \widehat{P}^k(s_h^k, a_h^k) \right) } + \mathbb{H}^2 \left( P^\star(s_h^k, a_h^k) \parallel \widehat{P}^k(s_h^k, a_h^k) \right) \Big)
$$

where we denote $V_{h+1;\widehat{P}}^{\pi^\star}(s' \sim P^\star(s,a))$ as the distribution of the random variable $V_{h+1;\widehat{P}}^{\pi^\star}(s')$ with $s' \sim P^\star(s,a)$. This is the key lemma used by Wang et al. (2024a) to show distributional RL can achieve second-order bounds. We show that this is also crucial for achieving a horizon-free bound.

Then, summing up over $k, h$, with Cauchy-Schwartz and the MLE generalization bound via Eluder dimension in Equation 4, we have

$$
\sum_{k \in \check{\mathcal{K}}} \sum_h \left| \mathbb{E}_{s' \sim \widehat{P}^k(s_h^k, a_h^k)} V_{h+1;\widehat{P}^k}^{\pi^k}(s') - \mathbb{E}_{s' \sim P^\star(s_h^k, a_h^k)} V_{h+1;\widehat{P}^k}^{\pi^k}(s') \right| \leq O\Big( \sum_{k \in \check{\mathcal{K}}} \sum_h \mathbb{H}^2 \left( P^\star(s_h^k, a_h^k) \parallel \widehat{P}^k(s_h^k, a_h^k) \right)
$$

$$
+ \sqrt{ \sum_{k \in \check{\mathcal{K}}} \sum_h \left( \mathbb{V}_{P^\star} V_{h+1;\widehat{P}^k}^{\pi^k} \right)(s_h^k, a_h^k) \sum_{k \in \check{\mathcal{K}}} \sum_h \mathbb{H}^2 \left( P^\star(s_h^k, a_h^k) \parallel \widehat{P}^k(s_h^k, a_h^k) \right) } \Big)
$$

$$
\leq O\Big( \sqrt{ \sum_{k \in \check{\mathcal{K}}} \sum_h \left( \mathbb{V}_{P^\star} V_{h+1;\widehat{P}^k}^{\pi^k} \right)(s_h^k, a_h^k) d_{\mathsf{RL}} \log(K |\mathcal{P}| / \delta) \log(KH) } + d_{\mathsf{RL}} \log(K |\mathcal{P}| / \delta) \log(KH) \Big). \tag{7}
$$

Note that we have $\left( \mathbb{V}_{P^\star} V_{h+1;\widehat{P}}^{\pi^k} \right)(s_h^k, a_h^k)$ depending on $\widehat{P}^k$. To get a second-order bound, we need to convert it to the variance under ground truth transition $P^\star$, and we want to do it without incurring any $H$ dependence. *This is another key difference from Wang et al. (2024a).*

We aim to replace $\left( \mathbb{V}_{P^\star} V_{h+1;\widehat{P}}^{\pi^k} \right)(s_h^k, a_h^k)$ by $\left( \mathbb{V}_{P^\star} V_{h+1}^{\pi^k} \right)(s_h^k, a_h^k)$ which is the variance under $P^\star$ (recall that $V^\pi$ is the value function of $\pi$ under $P^\star$), and we want to control the difference $\left( \mathbb{V}_{P^\star} \left( V_{h+1;\widehat{P}}^{\pi^k} - V_{h+1}^{\pi^k} \right) \right)(s_h^k, a_h^k)$. To do so, we need to bound the $2^m$ moment of the difference $V_{h+1;\widehat{P}}^{\pi^k} - V_{h+1}^{\pi^k}$ following the strategy in Zhang et al. (2021a); Zhou & Gu (2022); Zhao et al. (2023b). Let us define the following terms:

$$
A := \sum_{k \in \check{\mathcal{K}}} \sum_h \left[ \left( \mathbb{V}_{P^\star} V_{h+1;\widehat{P}^k}^{\pi^k} \right)(s_h^k, a_h^k) \right], \quad C_m := \sum_{k \in \check{\mathcal{K}}} \sum_h \left[ \left( \mathbb{V}_{P^\star} (V_{h+1;\widehat{P}^k}^{\pi^k} - V_{h+1}^{\pi^k})^{2^m} \right)(s_h^k, a_h^k) \right],
$$

$$
B := \sum_{k \in \check{\mathcal{K}}} \sum_h \left[ \left( \mathbb{V}_{P^\star} V_{h+1}^{\pi^k} \right)(s_h^k, a_h^k) \right], \quad G := \sqrt{ A \cdot d_{\mathsf{RL}} \log\left( \frac{K |\mathcal{P}|}{\delta} \right) \log(KH) } + d_{\mathsf{RL}} \log\left( \frac{K |\mathcal{P}|}{\delta} \right) \log(KH) .
$$

With the fact $\mathbb{V}_{P^\star}(a + b) \leq 2\mathbb{V}_{P^\star}(a) + 2\mathbb{V}_{P^\star}(b)$ we have $A \leq 2B + 2C_0$. For $C_m$, we prove that w.h.p. it has the recursive form $C_m \lesssim 2^m G + \sqrt{\log(1/\delta) C_{m+1}} + \log(1/\delta)$, during which process we also leverage the above Equation 7 and some careful analysis (detailed in Appendix D.1). Then, with the recursion lemma (Lemma 11), we can get $C_0 \lesssim G$, which further gives us

$$
A \lesssim B + d_{\mathsf{RL}} \log\left( \frac{K |\mathcal{P}|}{\delta} \right) \log(KH) + \sqrt{ A \cdot d_{\mathsf{RL}} \log\left( \frac{K |\mathcal{P}|}{\delta} \right) \log(KH) } \leq O\Big( B + d_{\mathsf{RL}} \log\left( \frac{K |\mathcal{P}|}{\delta} \right) \log(KH) \Big),
$$

where in the last step we use the fact $x \leq 2a + b^2$ if $x \leq a + b\sqrt{x}$. Finally, we note that $B \leq O(\sum_k \mathrm{VaR}_{\pi^k} + \log(1/\delta))$ w.h.p.. Plugging the upper bound of $A$ back into Equation 7 and then to Equation 6, we conclude the proof.

## 5 OFFLINE SETTING

For the offline setting, we directly analyze the Constrained Pessimism Policy Optimization (CPPO-LR) algorithm (Algorithm 2) proposed by Uehara & Sun (2021). We first explain the algorithm and then present its performance gap guarantee in finding the comparator policy $\pi^*$.

---

**Algorithm 2** (Uehara & Sun (2021)) Constrained Pessimistic Policy Optimization with Likelihood-Ratio based constraints (CPPO-LR)

1: **Input:** dataset $\mathcal{D} = \{s, a, s'\}$, model class $\mathcal{P}$, policy class $\Pi$, confidence parameter $\delta \in (0, 1)$, threshold $\beta$.
2: Calculate the confidence set based on the offline dataset:

$$\widehat{\mathcal{P}} = \left\{ P \in \mathcal{P} : \sum_{i=1}^{n} \log P(s_i'|s_i, a_i) \geq \max_{\tilde{P} \in \mathcal{P}} \sum_{i=1}^{n} \log \tilde{P}(s_i'|s_i, a_i) - \beta \right\}.$$

3: **Output:** $\hat{\pi} \leftarrow \operatorname{argmax}_{\pi \in \Pi} \min_{P \in \widehat{\mathcal{P}}} V_{0;P}^{\pi}(s_0)$.

---

Algorithm 2 splits the offline trajectory data that contains $K$ trajectories into a dataset of $(s, a, s')$ tuples (note that in total we have $n := KH$ many tuples) which is used to perform maximum likelihood estimation $\max_{\tilde{P} \in \mathcal{P}} \sum_{i=1}^{n} \log \tilde{P}(s_i'|s_i, a_i)$. It then builds a version space $\widehat{\mathcal{P}}$ which contains models $P \in \mathcal{P}$ whose log data likelihood is not below by too much than that of the MLE estimator. The threshold for the version space is constructed so that with high probability, $P^\star \in \widehat{\mathcal{P}}$. Once we build a version space, we perform pessimistic planning to compute $\hat{\pi}$.

We first define the single policy coverage condition as follows.

**Definition 3** (Single policy coverage). *Given any comparator policy $\pi^*$, denote the data-dependent single policy concentrability coefficient $C_{\mathcal{D}}^{\pi^*}$ as follows:*

$$C_{\mathcal{D}}^{\pi^*} := \max_{h, P \in \mathcal{P}} \frac{\mathbb{E}_{s,a \sim d_h^{\pi^*}} \mathbb{H}^2 \left( P(s, a) \parallel P^\star(s, a) \right)}{1/K \sum_{k=1}^{K} \mathbb{H}^2 \left( P(s_h^k, a_h^k) \parallel P^\star(s_h^k, a_h^k) \right)}.$$

*We assume w.p. at least $1 - \delta$ over the randomness of the generation of $\mathcal{D}$, we have $C_{\mathcal{D}}^{\pi^*} \leq C^{\pi^*}$.*

The existence of $C^{\pi^*}$ is certainly an assumption. We now give an example in the tabular MDP where we show that if the data is generated from some fixed behavior policy $\pi^b$ which has non-trivial probability of visiting every state-action pair, then we can show the existence of $C^{\pi^*}$.

**Example 2** (Tabular MDP with good behavior policy coverage). *If the $K$ trajectories are collected i.i.d. with a fixed behavior policy $\pi^b$, and $d_h^{\pi^b}(s, a) \geq \rho_{\min}, \forall s, a, h$ (similar to Ren et al. (2021)), then we have: if $K$ is large enough, i.e., $K \geq 2 \log(|\mathcal{S}||\mathcal{A}|H)/\rho_{\min}^2$, w.p. at least $1 - \delta$, $C_{\mathcal{D}}^{\pi^*} \leq 2/\rho_{\min}$.*

Our coverage definition (Definition 3) shares similar spirits as the one in Ye et al. (2024). It reflects how well the state-action samples in the offline dataset $\mathcal{D}$ cover the state-action pairs induced by the comparator policy $\pi^\star$. It is different from the coverage definition in Uehara & Sun (2021) in which the denominator is $\mathbb{E}_{s,a \sim d_h^{\pi^b}} \mathbb{H}^2 (P(s, a) \parallel P^\star(s, a))$ where $\pi^b$ is the fixed behavior policy used to collect $\mathcal{D}$. This definition does not apply in our setting since $\mathcal{D}$ is not necessarily generated by some underlying fixed behavior policy. On the other hand, our horizon-free result does not hold in the setting of Uehara & Sun (2021) where $\mathcal{D}$ is collected with a fixed behavior policy $\pi^b$ with the concentrability coefficient defined in their way. We leave the derivation of horizon-free results in the setting from Uehara & Sun (2021) as a future work.

Now we are ready to present the main theorem of Algorithm 2 for finite $\mathcal{P}$, which provides a tighter performance gap than that by Uehara & Sun (2021).

**Theorem 2** (Performance gap of Algorithm 2 with finite $\mathcal{P}$). *For any $\delta \in (0, 1)$, let $\beta = 4 \log(|\mathcal{P}|/\delta)$, w.p. at least $1 - \delta$, Algorithm 2 learns a policy $\hat{\pi}$ that enjoys the following performance gap with respect to any comparator policy $\pi^*$:*

$$V^{\pi^*} - V^{\hat{\pi}} \leq O \left( \sqrt{C^{\pi^*} \operatorname{VaR}_{\pi^*} \log(|\mathcal{P}|/\delta)/K} + C^{\pi^*} \log(|\mathcal{P}|/\delta)/K \right).$$

Comparing to the theorem (Theorem 2) of CPPO-LR from Uehara & Sun (2021), our bound has two improvements. First, our bound is horizon-free (not even any $\log(H)$ dependence), while the bound in Uehara & Sun (2021) has poly($H$) dependence. Second, our bound scales with $\operatorname{VaR}_{\pi^*} \in [0, 1]$, which can be small when $\operatorname{VaR}_{\pi^*} \ll 1$. For deterministic system and policy $\pi^*$, we have $\operatorname{VaR}_{\pi^*} = 0$ which means the sample complexity now scales at a faster rate $C^{\pi^*}/K$. The proof is in Appendix E.1.

We show that the same algorithm can achieve $1/K$ rate when $P^\star$ is deterministic (but rewards could be random, and the algorithm does not need to know the condition that $P^\star$ is deterministic).

**Corollary 4** ($C^{\pi^*}/K$ performance gap of Algorithm 2 with deterministic transitions). *When the ground truth transition $P^\star$ of the MDP is deterministic, for any $\delta \in (0,1)$, let $\beta = 4\log(|\mathcal{P}|/\delta)$, w.p. at least $1 - \delta$, Algorithm 2 learns a policy $\widehat{\pi}$ that enjoys the following performance gap with respect to any comparator policy $\pi^*$:*

$$V^{\pi^*} - V^{\widehat{\pi}} \leq O\left(C^{\pi^*}\log(|\mathcal{P}|/\delta)/K\right).$$

For infinite model class $\mathcal{P}$, we have a similar result in the following corollary.

**Corollary 5** (Performance gap of Algorithm 2 with infinite model class $\mathcal{P}$). *When the model class $\mathcal{P}$ is infinite, for any $\delta \in (0,1)$, let $\beta = 7\log(\mathcal{N}_{[]}((KH|\mathcal{S}|)^{-1}, \mathcal{P}, \|\cdot\|_\infty)/\delta)$, w.p. at least $1 - \delta$, Algorithm 2 learns a policy $\widehat{\pi}$ that enjoys the following PAC bound w.r.t. any comparator policy $\pi^*$:*

$$V^{\pi^*} - V^{\widehat{\pi}} \leq O\left(\sqrt{\frac{C^{\pi^*}\mathrm{VaR}_{\pi^*}\log(\mathcal{N}_{[]}((KH|\mathcal{S}|)^{-1}, \mathcal{P}, \|\cdot\|_\infty)/\delta)}{K}} + \frac{C^{\pi^*}\log(\mathcal{N}_{[]}((KH|\mathcal{S}|)^{-1}, \mathcal{P}, \|\cdot\|_\infty)/\delta)}{K}\right),$$

*where $\mathcal{N}_{[]}((KH|\mathcal{S}|)^{-1}, \mathcal{P}, \|\cdot\|_\infty)$ is the bracketing number defined in Definition 2.*

Our next example gives the explicit performance gap bound for tabular MDPs.

**Example 3** (Tabular MDPs). *For tabular MDPs, we have $\mathcal{N}_{[]}(\epsilon, \mathcal{P}, \|\cdot\|_\infty)$ upper-bounded by $(c/\epsilon)^{|\mathcal{S}|^2|\mathcal{A}|}$ (e.g., see Uehara & Sun (2021)). Then with probability at least $1 - \delta$, let $\beta = 7\log(\mathcal{N}_{[]}((KH|\mathcal{S}|)^{-1}, \mathcal{P}, \|\cdot\|_\infty)/\delta)$, Algorithm 2 learns a policy $\widehat{\pi}$ satisfying the following performance gap with respect to any comparator policy $\pi^*$:*

$$V^{\pi^*} - V^{\widehat{\pi}} \leq O\left(|\mathcal{S}|\sqrt{|\mathcal{A}|C^{\pi^*}\mathrm{VaR}_{\pi^*}\log(KH|\mathcal{S}|/\delta)/K} + |\mathcal{S}|^2|\mathcal{A}|C^{\pi^*}\log(KH|\mathcal{S}|/\delta)/K\right),$$

The closest result to us is from Ren et al. (2021), which analyzes the MBRL for tabular MDPs and obtains a performance gap $\tilde{O}(\sqrt{\frac{1}{Kd_m}} + \frac{|\mathcal{S}|}{Kd_m})$, where $d_m$ is the minimum visiting probability for the behavior policy to visit each state and action. Note that their result is not instance-dependent, which makes their gap only $\tilde{O}(1/\sqrt{K})$ even when the environment is deterministic and $\pi^*$ is deterministic. In a sharp contrast, our analysis shows a better $\tilde{O}(1/K)$ gap under the deterministic environment. Our result would still have the $\log H$ dependence, and we leave getting rid of this logarithmic dependence on the horizon $H$ as an open problem.

## 6 CONCLUSION

In this work, we presented a minimalist approach for achieving nearly horizon-free and second-order bounds in online and offline RL: simply train transition models via Maximum Likelihood Estimation followed by optimistic or pessimistic planning, depending on whether we operate in the online or offline learning mode. Our horizon-free bounds for function approximation look quite similar to the bounds in Contextual bandits, indicating that the need for long-horizon planning does not make RL harder than CB from a statistical perspective.

Our work has some limitations. First, though our results can achieve *completely horizon free* (not even $\log H$ dependence) for offline RL with finite model classes, in other settings we have $\log H$ dependence. We conjecture that the $\log(H)$ can be eliminated by using more careful analyses in dealing with the learned models' extrapolation errors via eluder dimensions, and using techniques such as peeling/chaining (Dudley, 1978; Zhang, 2006) when tackling infinite model classes. We leave this as an important future direction. Second, our model-based framework cannot capture problems that need to be solved via model-free approaches such as linear MDPs (Jin et al., 2020). An interesting future work is to see if we can extend our analysis to incorporate the function classes with small distributional Eduler dimensions, and if we can develop model-free approaches that can achieve horizon-free and instance-dependent bounds for RL with general function approximation. Finally, the algorithms studied in this work are not computationally tractable. This is due to the need of performing optimism/pessimism planning for exploration. Deriving computationally tractable RL algorithms for the rich function approximation setting is a long-standing question.

## 7 ACKNOWLEDGEMENT

Wen Sun acknowledges funding support from NSF IIS-2154711, NSF CAREER 2339395, DARPA LANCER: LeArning Network CybERagents. The work of John C.S. Lui was supported in part by the RGC GRF-14202923.

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

## A  SUMMARY OF CONTENTS IN THE APPENDIX

The Appendix is organized as follows.

In Appendix B, we provide some new analyses for Eluder dimension, which we will use for proving the regret bounds for the online RL setting.

In Appendix C, we provide some other supporting lemmas that will be used in our proofs.

In Appendix D, we provide the detailed proofs for the online RL setting (Section 4). Specifically, in Appendix D.1 we give the proof of Theorem 1; in Appendix D.2, we show the proof of Corollary 2; in Appendix D.3, we give the proof of Corollary 3.

In Appendix E, we provide the detailed proofs for the offline RL setting (Section 5). Specifically, in Appendix E.1 we give the proof of Theorem 2; in Appendix E.2, we show the proof of Corollary 4; in Appendix E.3, we give the proof of Corollary 5; in Appendix E.4, we show the proof of the claim in Example 2.

## B  ANALYSIS REGARDING THE ELUDER DIMENSION

For simplicity, we denote $x_h^k = (s_h^k, a_h^k)$.

First we have two technical lemma. The first lemma bounds the summation of "self-normalization" terms by the Eluder dimension. Our result generalizes the previous result by Zhao et al. (2023a) from the $\ell_2$-Eluder dimension to the $\ell_1$ case.

**Lemma 3.** *Suppose for all $g \in \Psi, |g| \le 1$ and $\lambda > 1$, then we have*

$$\sum_{k=1}^{K} \sum_{h=1}^{H} \min \left\{ 1, \sup_{g \in \Psi} \frac{|g(x_h^k)|}{\sum_{k'=1}^{k-1} \sum_{h'=1}^{H} |g(x_{h'}^{k'})| + \sum_{h'=1}^{h-1} |g(x_{h'}^k)| + \lambda} \right\}$$
$$\le 12 \log^2 (4\lambda KH) \cdot DE_1(\Psi, \mathcal{S} \times \mathcal{A}, 1/(8\lambda KH)) + \lambda^{-1}.$$

*Proof of Lemma 3.* We follow the proof steps of Theorem 4.6 in Zhao et al. (2023a). For simplicity, we use $n = KH$, $i = kH + h$ to denote the indices and denote $x_h^k$ by $x_i$. Then we need to prove

$$\sum_{i=1}^{n} \min \left\{ 1, \sup_{g \in \Psi} \frac{|g(x_i)|}{\sum_{t=1}^{i-1} |g(x_t)| + \lambda} \right\} \le 12 \log^2 (4\lambda n) \cdot DE_1(\Psi, \mathcal{S} \times \mathcal{A}, 1/(8\lambda n)) + \lambda^{-1} \quad (8)$$

Let

$$g_i = \operatorname*{argmax}_{g \in \Psi} \frac{|g(x_i)|}{\sum_{j=1}^{i-1} |g(x_j)| + \lambda} \quad (9)$$

For any $1/(\lambda n) \le \rho \le 1$ and $1 \le j \le \lceil \log(4\lambda n) \rceil$, we define

$$A_\rho^j = \left\{ i \in [n] : 2^{-j} < |g_i(x_i)| \le 2^{-j+1}, \frac{|g_i(x_i)|}{\sum_{t=1}^{i-1} |g_i(x_t)| + \lambda} \ge \rho/2 \right\}, \quad d_j := DE_1(\Psi, \mathcal{S} \times \mathcal{A}, 2^{-j}). \quad (10)$$

Next we only consider the set $A_\rho^j$ where $|A_\rho^j| > d_j$. We denote $A_\rho^j = \{a_1, \dots, a_A\}$, where $A = |A_\rho^j|$ and $\{a_i\}$ keeps the same order as $\{x_i\}$. Next we do the following constructions. We maintain $k = \lfloor (A-1)/d_j \rfloor$ number of queues $Q_1, \dots, Q_k$, all of them initialized as emptysets. We put $a_1$ into $Q_1$. For $a_i, i \ge 2$, we put $a_i$ into $Q_l$, where $Q_l$ is the first queue where $a_i$ is $2^{-j}$-independent of all elements in $Q_l$. Let $i_{\max}$ be the smallest $i$ when we can not put $a_i$ into any existing queue.

We claim that $i_{\max}$ indeed exists, i.e., our construction will stop before we put all elements in $A_\rho^j$ into $Q_1, \dots, Q_k$. In fact, note the fact that the length of each $Q_l$ is always no more than $d_j$, which is due to the fact that any $2^{-j}$-independent sequence's length is at most $d_j$. Meanwhile, since we only have $k = \lfloor (A-1)/d_j \rfloor$, then the amount of elements in $Q_1 \cup \dots \cup Q_k$ will be upper bounded by $k \cdot d_j < A$. That suggests at least one element in $A_\rho^j$ is not contained by $Q_1 \cup \dots \cup Q_k$, i.e., $i_{\max}$ exists.

By the definition of $i_{\max}$, we know that $a_{i_{\max}}$ is $2^{-j}$-dependent to each $Q_l$. Next we give a bound of $A$. First, note

$$\sum_{t=1}^{i_{\max}-1} |g_{i_{\max}}(x_t)| \ge \sum_{t \in Q_1 \cup \dots \cup Q_k} |g_{i_{\max}}(a_t)| = \sum_{l=1}^{k} \sum_{t \in Q_l} |g_{i_{\max}}(a_t)| > k \cdot 2^{-j}, \quad (11)$$

where the first inequality holds since $Q_l$ are the elements that appear before $a_{i_{\max}}$, the second one holds due to the following induction of Eluder dimension: since $a_{i_{\max}}$ is $2^{-j}$-dependent to $Q_l$, then we have

$$\forall g \in \Psi, \sum_{t \in Q_l} |g(a_t)| \le 2^{-j} \Rightarrow |g(a_{i_{\max}})| \le 2^{-j}. \tag{12}$$

Therefore, given the fact $|g_{i_{\max}}(a_{i_{\max}})| > 2^{-j}$ (recall the definition of $A_\rho^j$), we must have $\sum_{t \in Q_l} |g_{i_{\max}}(a_t)| > 2^{-j}$ as well, which suggests the second inequality of Equation 11 holds. Second, we have

$$\sum_{t=1}^{i_{\max}-1} |g_{i_{\max}}(x_t)| \le 2/\rho \cdot |g_{i_{\max}}(a_{i_{\max}})| \le 4 \cdot 2^{-j}/\rho, \tag{13}$$

where both inequalities hold due to the definition of $A_\rho^j$. Combining Equation 11 and Equation 13, we have

$$k < 4/\rho \Rightarrow A \le 4d_j/\rho + d_j \le 5d_j/\rho. \tag{14}$$

Therefore, we have that for all $\rho, j$, $|A_\rho^j| \le 5d_j/\rho$.

Finally we prove Equation 8. $1/(\lambda n) \le \rho \le 1$ and $1 \le j \le \lceil \log(4\lambda n) \rceil = J$. Denote

$$A_\rho = \left\{ i \in [n] : \frac{|g_i(x_i)|}{\sum_{t=1}^{i-1} |g_i(x_t)| + \lambda} \ge \rho/2 \right\}. \tag{15}$$

Then it is easy to notice that $|A_\rho| = \sum_j |A_\rho^j| \le \lceil \log(4\lambda n) \rceil \cdot 5d_J/\rho$, where we use the fact that the Eluder dimension $d_j$ is increasing. Therefore, by the standard peeling technique, we have

$$\begin{aligned}
\sum_{i=1}^n \min\left\{1, \sup_{g \in \Psi} \frac{|g(x_i)|}{\sum_{t=1}^{i-1} |g(x_t)| + \lambda}\right\} &= \sum_{j \in [\lceil \log(\lambda n) \rceil]} \sum_{i \in A_{2^{-j}} \setminus A_{2^{-j+1}}} + \sum_{j = \lceil \log(\lambda n) \rceil} \sum_{i \notin A_{2^{-j+1}}} \\
&\le \sum_{j \in [\lceil \log(\lambda n) \rceil]} \sum_{i \in A_{2^{-j}} \setminus A_{2^{-j+1}}} 2^{-j-1} + n \cdot 1/(\lambda n) \\
&\le \sum_{j \in [\lceil \log(\lambda n) \rceil]} \sum_{i \in A_{2^{-j}}} 2^{-j-1} + n \cdot 1/(\lambda n) \\
&\le \lceil \log(\lambda n) \rceil \cdot \lceil \log(4\lambda n) \rceil \cdot 3d_J + \lambda^{-1},
\end{aligned}$$

which concludes our proof. $\qquad\square$

Next lemma gives a bound to bound the number of episodes where the behavior along these episodes are "bad". Intuitively speaking, our lemma suggests we only have limited number of bad episodes, therefore won't affect the final performance of our algorithm.

**Lemma 4.** *Given $\lambda > 1$. There exists at most*

$$13 \log^2(4\lambda KH) \cdot DE_1(\Psi, \mathcal{S} \times \mathcal{A}, 1/(8\lambda KH)) \tag{16}$$

*number of $k \in [K]$ satisfying the following claim*

$$\sup_{g \in \Psi} \frac{\lambda + \sum_{k'=1}^k \sum_{h'=1}^H |g(x_{h'}^{k'})|}{\lambda + \sum_{k'=1}^{k-1} \sum_{h'=1}^H |g(x_{h'}^{k'})|} > 4. \tag{17}$$

*Proof of Lemma 4.* Note that

$$\begin{aligned}
\sum_{k=1}^K \min\left\{2, \log \sup_{g \in \Psi} \frac{\lambda + \sum_{k'=1}^k \sum_{h'=1}^H |g(x_{h'}^{k'})|}{\lambda + \sum_{k'=1}^{k-1} \sum_{h'=1}^H |g(x_{h'}^{k'})|}\right\} \\
\le \sum_{k=1}^K \min\left\{2, \log \prod_{h=1}^H \sup_{g \in \Psi} \frac{\lambda + \sum_{k'=1}^{k-1} \sum_{h'=1}^H |g(x_{h'}^{k'})| + \sum_{h'=1}^h |g(x_{h'}^k)|}{\lambda + \sum_{k'=1}^{k-1} \sum_{h'=1}^H |g(x_{h'}^{k'})| + \sum_{h'=1}^{h-1} |g(x_{h'}^k)|}\right\} \\
= \sum_{k=1}^K \min\left\{2, \sum_{h=1}^H \log\left(1 + \sup_{g \in \Psi} \frac{|g(x_h^k)|}{\lambda + \sum_{k'=1}^{k-1} \sum_{h'=1}^H |g(x_{h'}^{k'})| + \sum_{h'=1}^{h-1} |g(x_{h'}^k)|}\right)\right\}
\end{aligned}$$

$$\leq \sum_{k=1}^{K} \sum_{h=1}^{H} \min\left\{2, \sup_{g\in\Psi} \frac{|g(x_h^k)|}{\sum_{k'=1}^{k-1}\sum_{h'=1}^{H}|g(x_{h'}^{k'})| + \sum_{h'=1}^{h-1}|g(x_{h'}^k)| + \lambda}\right\},$$

$$\leq 2\sum_{k=1}^{K}\sum_{h=1}^{H}\min\left\{1, \sup_{g\in\Psi} \frac{|g(x_h^k)|}{\sum_{k'=1}^{k-1}\sum_{h'=1}^{H}|g(x_{h'}^{k'})| + \sum_{h'=1}^{h-1}|g(x_{h'}^k)| + \lambda}\right\}$$

$$\leq 24\log^2(4\lambda KH) \cdot DE_1(\Psi, \mathcal{S}\times\mathcal{A}, 1/(8\lambda KH)) + 2\lambda^{-1}$$

$$\leq 26\log^2(4\lambda KH) \cdot DE_1(\Psi, \mathcal{S}\times\mathcal{A}, 1/(8\lambda KH)). \tag{18}$$

where the first inequality holds since $\sup_g \prod f(g) \leq \prod \sup_g f(g)$, the second one holds since $\log(1+x) \leq x$, the fourth one holds due to Lemma 3. Therefore, there are at most

$$26\log^2(4\lambda KH) \cdot DE_1(\Psi, \mathcal{S}\times\mathcal{A}, 1/(8\lambda KH))/2$$

number of $k$ satisfying

$$\log\sup_{g\in\Psi} \frac{\lambda + \sum_{k'=1}^{k}\sum_{h'=1}^{H}|g(x_{h'}^{k'})|}{\lambda + \sum_{k'=1}^{k-1}\sum_{h'=1}^{H}|g(x_{h'}^{k'})|} > 2,$$

which concludes the proof.

$$\square$$

We next have the following lemma, which bounds the regret by the Eluder dimension.

**Lemma 5** (Theorem 5.3, Wang et al. 2023). *Let $C := \sup_{(s,a)\in\mathcal{S}\times\mathcal{A}, f\in\Psi}|f((s,a))|$ be the envelope. For any sequences $f^{(1)}, \ldots, f^{(N)} \subseteq \Psi$, $(s,a)^{(1)}, \ldots, (s,a)^{(N)} \subseteq \mathcal{S}\times\mathcal{A}$, let $\beta$ be a constant such that for all $n\in[N]$ we have, $\sum_{i=1}^{n-1}|f^{(n)}((s,a)^i)| \leq \beta$. Then, for all $n\in[N]$, we have*

$$\sum_{t=1}^{n}|f^{(t)}((s,a)^t)| \leq \inf_{0<\epsilon\leq 1}\left\{DE_1(\Psi, \mathcal{S}\times\mathcal{A}, \epsilon)(2C + \beta\log(C/\epsilon)) + n\epsilon\right\}.$$

Given Lemma 4 and Lemma Lemma 5, we are able to prove the following key lemma.

**Lemma 6** (New Eluder Pigeon Lemma). *Let the event $\mathcal{E}$ be*

$$\mathcal{E}: \forall k\in[K], \sum_{i=1}^{k-1}\sum_{h=1}^{H}\mathbb{H}^2(\widehat{P}^k(s_h^i, a_h^i)\|P^*(s_h^i, a_h^i)) \leq \eta. \tag{19}$$

*Then under event $\mathcal{E}$, there exists a set $\mathcal{K}\in[K]$ such that*

- *We have $|\mathcal{K}| \leq 13\log^2(4\eta KH) \cdot DE_1(\Psi, \mathcal{S}\times\mathcal{A}, 1/(8\eta KH))$.*

- *We have*

$$\sum_{k\in[K]\setminus\mathcal{K}}\sum_{h=1}^{H}\mathbb{H}^2\left(P^\star(s_h^k, a_h^k) \,\|\, \widehat{P}^k(s_h^k, a_h^k)\right)$$
$$\leq \inf_{0<\epsilon\leq 1}\left\{DE_1(\Psi, \mathcal{S}\times\mathcal{A}, \epsilon)(2 + 7\eta\log(1/\epsilon)) + KH\epsilon\right\}$$
$$\leq DE_1(\Psi, \mathcal{S}\times\mathcal{A}, 1/KH)(2 + 7\eta\log(KH)) + 1, \tag{20}$$

*where the function class $\Psi = \{(s,a) \mapsto \mathbb{H}^2(P^\star(s,a) \,\|\, P(s,a)): P\in\mathcal{P}\}$.*

*Proof of Lemma 6.* We interchangeably use $n = kH + h$ to denote the indices of $s_h^k, a_h^k$. We set $f^{(n)}((s,a))$ in Lemma 5 as $H^2(P^k(s,a)\|P^*(s,a))$.

First, we prove that the $\beta$ in Lemma 5 can be selected as $7\eta$ under event $\mathcal{E}$. To show that, let $\mathcal{K}$ denote all the $k$ stated in Lemma 4. Then for all $k$ such that $k+1 \notin \mathcal{K}$, $h = 2, \ldots, H$, let $n = kH + h$, we have

$$\sum_{i=0}^{n-1}|f^{(n)}((s,a)^i)| \leq \sum_{i=0}^{kH+H}|f^{(n)}((s,a)^i)|$$

$$= \left( \lambda + \sum_{i=0}^{kH} |f^{(n)}((s,a)^i)| \right) \cdot \frac{\sum_{i=0}^{kH+H} |f^{(n)}((s,a)^i)| + \lambda}{\sum_{i=0}^{kH} |f^{(n)}((s,a)^i)| + \lambda} - \lambda$$

$$\le \left( \lambda + \sum_{i=0}^{kH} |f^{(n)}((s,a)^i)| \right) \cdot 4 - \lambda$$

$$\le 7\eta, \tag{21}$$

where the second inequality holds due to Lemma 4, the last one holds due to the definition of $\mathcal{E}$. Therefore, we prove our lemma by the conclusion of Lemma 5 with $\beta = 7\eta$. $\qquad\square$

## C  OTHER SUPPORTING LEMMAS

**Lemma 7** (Simulation Lemma (Agarwal et al. (2019))). *We have*

$$V_{0;P^\star}^\pi - V_{0;\hat{P}}^\pi \le \sum_{h=0}^{H-1} \mathbb{E}_{s,a\sim d_h^\pi} \left[ \left| \mathbb{E}_{s'\sim P^\star(s,a)} V_{h+1;\hat{P}}^\pi(s') - \mathbb{E}_{s'\sim\hat{P}(s,a)} V_{h+1;\hat{P}}^\pi(s') \right| \right].$$

**Lemma 8** (Change of Variance Lemma (Lemma C.5 in Jin et al. (2018))).

$$\sum_{h=0}^{H-1} \mathbb{E}_{s,a\sim d_h^\pi} \left[ \left( \mathbb{V}_{P^\star} V_{h+1;P^\star}^\pi \right)(s,a) \right] = \mathrm{VaR}_\pi.$$

**Lemma 9** (Generalization bounds of MLE for finite model class (Theorem E.4 in Wang et al. (2023))). *Let $\mathcal{X}$ be the context/feature space and $\mathcal{Y}$ be the label space, and we are given a dataset $D = \{(x_i, y_i)\}_{i\in[n]}$ from a martingale process: for $i = 1, 2, ..., n$, sample $x_i \sim \mathcal{D}_i(x_{1:i-1}, y_{1:i-1})$ and $y_i \sim p(\cdot \mid x_i)$. Let $f^\star(x,y) = p(y \mid x)$ and we are given a realizable, i.e., $f^\star \in \mathcal{F}$, function class $\mathcal{F}: \mathcal{X}\times\mathcal{Y}\to\Delta(\mathbb{R})$ of distributions. Suppose $\mathcal{F}$ is finite. Fix any $\delta \in (0,1)$, set $\beta = \log(|\mathcal{F}|/\delta)$ and define*

$$\widehat{\mathcal{F}} = \left\{ f \in \mathcal{F} : \sum_{i=1}^n \log f(x_i, y_i) \ge \max_{\widetilde{f}\in\mathcal{F}} \sum_{i=1}^n \log \widetilde{f}(x_i, y_i) - 4\beta \right\}.$$

*Then w.p. at least $1 - \delta$, the following holds:*

*(1) The true distribution is in the version space, i.e., $f^\star \in \widehat{\mathcal{F}}$.*

*(2) Any function in the version space is close to the ground truth data-generating distribution, i.e., for all $f \in \widehat{\mathcal{F}}$*

$$\sum_{i=1}^n \mathbb{E}_{x\sim\mathcal{D}_i} \left[ \mathbb{H}^2(f(x,\cdot) \parallel f^\star(x,\cdot)) \right] \le 22\beta.$$

**Lemma 10** (Generalization bounds of MLE for infinite model class (Theorem E.5 in Wang et al. (2023))). *Let $\mathcal{X}$ be the context/feature space and $\mathcal{Y}$ be the label space, and we are given a dataset $D = \{(x_i, y_i)\}_{i\in[n]}$ from a martingale process: for $i = 1, 2, ..., n$, sample $x_i \sim \mathcal{D}_i(x_{1:i-1}, y_{1:i-1})$ and $y_i \sim p(\cdot \mid x_i)$. Let $f^\star(x,y) = p(y \mid x)$ and we are given a realizable, i.e., $f^\star \in \mathcal{F}$, function class $\mathcal{F}: \mathcal{X}\times\mathcal{Y}\to\Delta(\mathbb{R})$ of distributions. Suppose $\mathcal{F}$ is finite. Fix any $\delta \in (0,1)$, set $\beta = \log(\mathcal{N}_{[]}((n|\mathcal{Y}|)^{-1}, \mathcal{F}, \|\cdot\|_\infty)/\delta)$ (where $\mathcal{N}_{[]}((n|\mathcal{Y}|)^{-1}, \mathcal{F}, \|\cdot\|_\infty)$ is the bracketing number defined in Definition 2) and define*

$$\widehat{\mathcal{F}} = \left\{ f \in \mathcal{F} : \sum_{i=1}^n \log f(x_i, y_i) \ge \max_{\widetilde{f}\in\mathcal{F}} \sum_{i=1}^n \log \widetilde{f}(x_i, y_i) - 7\beta \right\}.$$

*Then w.p. at least $1 - \delta$, the following holds:*

*(1) The true distribution is in the version space, i.e., $f^\star \in \widehat{\mathcal{F}}$.*

*(2) Any function in the version space is close to the ground truth data-generating distribution, i.e., for all $f \in \widehat{\mathcal{F}}$*

$$\sum_{i=1}^n \mathbb{E}_{x\sim\mathcal{D}_i} \left[ \mathbb{H}^2(f(x,\cdot) \parallel f^\star(x,\cdot)) \right] \le 28\beta.$$

**Lemma 11** (Recursion Lemma). *Let $G > 0$ be a positive constant, $a < G/2$ is also a positive constant, and let $\{C_m\}_{m=0}^{N=\lceil \log_2(\frac{KH}{G}) \rceil}$ be a sequence of positive real numbers satisfying:*

1. $C_m \leq 2^m G + \sqrt{aC_{m+1}} + a$ *for all $m \geq 0$,*

2. $C_m \leq KH$ *for all $m \geq 0$, where $K > 0$ and $H > 0$ are positive constants.*

*Then, it holds that:*
$$C_0 \leq 4G.$$

*Proof of Lemma 11.* We will prove by induction that for all $m \geq 0$,
$$C_m \leq 2^{m+2}G.$$

Then, for $m = 0$, this would immediately show $C_0 \leq 4G$.

**1. The base case $m = N$:**

Since $N = \lceil \log_2(\frac{KH}{G}) \rceil$, it is obvious that $2^{N+2}G \geq KH$. Thus, $C_N \leq KH \leq 2^{N+2}G$, the inequality holds for $m = N$.

**2. The induction step:**

Assume that for some $m \geq 0$, for $C_{m+1}$, we have:
$$C_{m+1} \leq 2^{m+1+2}G = 2^{m+3}G.$$

Then, we have

$$
\begin{aligned}
C_m &\leq 2^m G + \sqrt{aC_{m+1}} + a \\
&\leq 2^m G + \sqrt{a2^{m+3}G} + a \\
&\leq 2^m G + \sqrt{\frac{G}{2} \cdot 2^{m+3}G} + \frac{G}{2} \\
&= G \cdot (2^m + 2^{m/2+1} + 2^{-1}) \\
&\leq G \cdot (2^m + 2^{m+1} + 2^m) \\
&= 2^{m+2}G .
\end{aligned}
\tag{22}
$$

Therefore, by induction, we have for all $m \geq 0$,
$$C_m \leq 2^{m+2}G.$$

And the proof follows by setting $m = 0$. $\qquad\square$

## D    DETAILED PROOFS FOR THE ONLINE SETTING IN SECTION 4

### D.1    PROOF OF THEOREM 1

The following is the full proof of Theorem 1.

For notational simplicity, throughout this whole section, we denote

$$A := \sum_{k \in [K-1] \setminus \mathcal{K}} \sum_{h=0}^{H-1} \left[ \left( \mathbb{V}_{P^\star} V_{h+1;\widehat{P}^k}^{\pi^k} \right)(s_h^k, a_h^k) \right]$$

$$B := \sum_{k \in [K-1] \setminus \mathcal{K}} \sum_{h=0}^{H-1} \left[ \left( \mathbb{V}_{P^\star} V_{h+1}^{\pi^k} \right)(s_h^k, a_h^k) \right],$$

$$C_m := \sum_{k \in [K-1] \setminus \mathcal{K}} \sum_{h=0}^{H-1} \left[ \left( \mathbb{V}_{P^\star} (V_{h+1;\widehat{P}^k}^{\pi^k} - V_{h+1}^{\pi^k})^{2^m} \right)(s_h^k, a_h^k) \right]$$

$$G := \sqrt{\sum_{k \in [K-1] \setminus \mathcal{K}} \sum_{h=0}^{H-1} \left[ \left( \mathbb{V}_{P^\star} V_{h+1;\widehat{P}^k}^{\pi^k} \right)(s_h^k, a_h^k) \right] \cdot \mathrm{DE}_1(\Psi, \mathcal{S} \times \mathcal{A}, 1/KH) \cdot \log(K |\mathcal{P}|/\delta) \log(KH)}$$
$$+ \mathrm{DE}_1(\Psi, \mathcal{S} \times \mathcal{A}, 1/KH) \cdot \log(K |\mathcal{P}|/\delta) \log(KH)$$
$$I_h^k := \mathbb{E}_{s' \sim P^\star(s_h^k, a_h^k)} V_{h+1;\widehat{P}^k}^{\pi^k}(s') - V_{h+1;\widehat{P}^k}^{\pi^k}(s_{h+1}^k) \tag{23}$$

We use $\mathbb{I}\{\cdot\}$ to denote the indicator function. We define the following events which we will later show that they happen with high probability.

$$\mathcal{E}_1 := \left\{ \forall k \in [K-1] : P^\star \in \widehat{\mathcal{P}}^k, \text{and} \sum_{i=0}^{k-1} \sum_{h=0}^{H-1} \mathbb{H}^2(P^\star(s_h^i, a_h^i) \| \widehat{P}^k(s_h^i, a_h^i)) \leq 22 \log(K |\mathcal{P}|/\delta). \right\},$$
$$\tag{24}$$

$$\mathcal{E}_2 := \left\{ \sum_{k \in [K-1] \setminus \mathcal{K}} \sum_{h=0}^{H-1} I_h^k \lesssim \sqrt{\sum_{k \in [K-1] \setminus \mathcal{K}} \sum_{h=0}^{H-1} \left( \mathbb{V}_{P^\star} V_{h+1;\widehat{P}^k}^{\pi^k} \right)(s_h^k, a_h^k) \log(1/\delta) + \log(1/\delta)} \right\}, \tag{25}$$

$$\mathcal{E}_3 := \mathcal{E}_1 \cap \left\{ \forall m \in [0, \lceil \log_2(\frac{KH}{G}) \rceil] : C_m \lesssim 2^m G + \sqrt{\log(1/\delta) \cdot C_{m+1}} + \log(1/\delta) \right\}, \tag{26}$$

$$\mathcal{E}_4 := \left\{ \sum_{k=0}^{K-1} \sum_{h=0}^{H-1} \left[ \left( \mathbb{V}_{P^\star} V_{h+1}^{\pi^k} \right)(s_h^k, a_h^k) \right] \lesssim \sum_{k=0}^{K-1} \mathrm{VaR}_{\pi^k} + \log(1/\delta) \right\}, \tag{27}$$

$$\mathcal{E}_5 := \left\{ \sum_{k=0}^{K-1} \sum_{h=1}^{H} r(s_h^k, a_h^k) - \sum_{k=0}^{K-1} V_{0;P^\star}^{\pi^k} \lesssim \sqrt{\sum_{k=0}^{K-1} \mathrm{VaR}_{\pi^k} \log(1/\delta)} + \log(1/\delta) \right\}, \tag{28}$$

$$\mathcal{E} := \mathcal{E}_2 \cap \mathcal{E}_3 \cap \mathcal{E}_4 \cap \mathcal{E}_5. \tag{29}$$

First, by the realizability assumption, the standard generalization bound for MLE (Lemma 9) with simply setting $D_i$ to be the delta distribution on the realized $(s_h^k, a_h^k)$ pairs, and a union bound over $K$ episodes, we have that w.p. at least $1 - \delta$, for any $k \in [0, K-1]$:

(1) $P^\star \in \widehat{\mathcal{P}}^k$;

(2)
$$\sum_{i=0}^{k-1} \sum_{h=0}^{H-1} \mathbb{H}^2(P^\star(s_h^i, a_h^i) \| \widehat{P}^k(s_h^i, a_h^i)) \leq 22 \log(K |\mathcal{P}|/\delta). \tag{30}$$

This directly indicates that
$$P(\mathbb{I}\{\mathcal{E}_1\}) \geq 1 - \delta. \tag{31}$$

Under event $\mathcal{E}_1$, with the realizability in above (1), and by the optimistic algorithm design $(\pi^k, \widehat{P}^k) \leftarrow \mathrm{argmax}_{\pi \in \Pi, P \in \widehat{\mathcal{P}}^k} V_{0;P}^\pi(s_0)$, for any $k \in [0, K-1]$, we have the following optimism guarantee

$$V_{0;P^\star}^\star \leq \max_{\pi \in \Pi, P \in \widehat{\mathcal{P}}^k} V_{0;P}^\pi = V_{0;\widehat{P}^k}^{\pi^k}.$$

Then, under event $\mathcal{E}_1$, we use Lemma 6 and Equation 30 to get the following:

There exists a set $\mathcal{K} \subseteq [K-1]$ such that

- $|\mathcal{K}| \leq 13 \log^2(88 \log(K |\mathcal{P}|/\delta) KH) \cdot DE_1(\Psi, \mathcal{S} \times \mathcal{A}, 1/(176 \log(K |\mathcal{P}|/\delta) KH))$
- And

$$\sum_{k \in [K-1] \setminus \mathcal{K}} \sum_{h=0}^{H-1} \mathbb{H}^2 \left( P^\star(s_h^k, a_h^k) \| \widehat{P}^k(s_h^k, a_h^k) \right)$$
$$\leq DE_1(\Psi, \mathcal{S} \times \mathcal{A}, 1/KH) \cdot (2 + 154 \log(K |\mathcal{P}|/\delta) \log(KH)) + 1$$
$$\lesssim DE_1(\Psi, \mathcal{S} \times \mathcal{A}, 1/KH) \cdot \log(K |\mathcal{P}|/\delta) \log(KH). \tag{32}$$

We upper bound the regret with optimism, and by dividing $k \in [K-1]$ into $\mathcal{K}$ and $[K-1] \setminus \mathcal{K}$ with the assumption that the trajectory-wise cumulative reward is normalized in $[0,1]$, as follows

$$
\sum_{k=0}^{K-1} V_{0;P^\star}^\star - \sum_{k=0}^{K-1} \sum_{h=1}^H r(s_h^k, a_h^k)
$$

$$
\leq |\mathcal{K}| + \sum_{k \in [K-1] \setminus \mathcal{K}} \left( V_{0;\widehat{P}^k}^{\pi^k} - \sum_{h=0}^{H-1} r(s_h^k, a_h^k) \right)
$$

$$
\lesssim \log^2(\log(K|\mathcal{P}|/\delta)KH) \cdot DE_1(\Psi, \mathcal{S} \times \mathcal{A}, 1/(\log(K|\mathcal{P}|/\delta)KH)) + \sum_{k \in [K-1] \setminus \mathcal{K}} \left( V_{0;\widehat{P}^k}^{\pi^k} - \sum_{h=0}^{H-1} r(s_h^k, a_h^k) \right).
$$

$$(33)$$

We then do the following decomposition. Note that for any $k \in [K-1]$, policy $\pi^k$ is deterministic. We have that for any $k \in [K-1]$

$$
V_{0;\widehat{P}^k}^{\pi^k}(s_0^k) - \sum_{h=0}^{H-1} r(s_h^k, a_h^k)
$$

$$
= Q_{0;\widehat{P}^k}^{\pi^k}(s_0^k, a_0^k) - \sum_{h=0}^{H-1} r(s_h^k, a_h^k)
$$

$$
= r(s_0^k, a_0^k) + \mathbb{E}_{s' \sim \widehat{P}^k(s_0^k, a_0^k)} V_{1;\widehat{P}^k}^{\pi^k}(s') - \sum_{h=0}^{H-1} r(s_h^k, a_h^k)
$$

$$
= \mathbb{E}_{s' \sim \widehat{P}^k(s_0^k, a_0^k)} V_{1;\widehat{P}^k}^{\pi^k}(s') - \sum_{h=1}^{H} r(s_h^k, a_h^k)
$$

$$
= \mathbb{E}_{s' \sim P^*(s_0^k, a_0^k)} V_{1;\widehat{P}^k}^{\pi^k}(s') - \sum_{h=1}^{H} r(s_h^k, a_h^k) + \mathbb{E}_{s' \sim \widehat{P}^k(s_0^k, a_0^k)} V_{1;\widehat{P}^k}^{\pi^k}(s') - \mathbb{E}_{s' \sim P^*(s_0^k, a_0^k)} V_{1;\widehat{P}^k}^{\pi^k}(s')
$$

$$
= V_{1;\widehat{P}^k}^{\pi^k}(s_1^k) - \sum_{h=1}^{H-1} r(s_h^k, a_h^k) + \underbrace{\mathbb{E}_{s' \sim P^*(s_0^k, a_0^k)} V_{1;\widehat{P}^k}^{\pi^k}(s') - V_{1;\widehat{P}^k}^{\pi^k}(s_1^k)}_{I_0^k}
$$

$$
+ \mathbb{E}_{s' \sim \widehat{P}^k(s_0^k, a_0^k)} V_{1;\widehat{P}^k}^{\pi^k}(s') - \mathbb{E}_{s' \sim P^*(s_0^k, a_0^k)} V_{1;\widehat{P}^k}^{\pi^k}(s'),
$$

where we use the Bellman equation for several times.

Then, by doing this recursively, we can get for any $k \in [K-1]$

$$
V_{0;\widehat{P}^k}^{\pi^k}(s_h^k) - \sum_{h=0}^{H-1} r(s_h^k, a_h^k)
$$

$$
\leq \sum_{h=0}^{H-1} I_h^k + \sum_{h=0}^{H-1} \left| \mathbb{E}_{s' \sim \widehat{P}^k(s_h^k, a_h^k)} V_{h+1;\widehat{P}^k}^{\pi^k}(s') - \mathbb{E}_{s' \sim P^*(s_h^k, a_h^k)} V_{h+1;\widehat{P}^k}^{\pi^k}(s') \right| \qquad (34)
$$

Therefore,

$$
\sum_{k \in [K-1] \setminus \mathcal{K}} \left( V_{0;\widehat{P}^k}^{\pi^k}(s_h^k) - \sum_{h=0}^{H-1} r(s_h^k, a_h^k) \right)
$$

$$
\leq \sum_{k \in [K-1] \setminus \mathcal{K}} \sum_{h=0}^{H-1} I_h^k + \sum_{k \in [K-1] \setminus \mathcal{K}} \sum_{h=0}^{H-1} \left| \mathbb{E}_{s' \sim \widehat{P}^k(s_h^k, a_h^k)} V_{h+1;\widehat{P}^k}^{\pi^k}(s') - \mathbb{E}_{s' \sim P^*(s_h^k, a_h^k)} V_{h+1;\widehat{P}^k}^{\pi^k}(s') \right| \quad (35)
$$

Next we bound $\sum_{k \in [K-1] \setminus \mathcal{K}} \sum_{h=0}^{H-1} I_h^k$. Note that by Azuma Bernstein's inequality, with probability at least $1 - \delta$

$$
\sum_{k \in [K-1] \setminus \mathcal{K}} \sum_{h=0}^{H-1} I_h^k \leq \sqrt{2 \sum_{k \in [K-1] \setminus \mathcal{K}} \sum_{h=0}^{H-1} \left( \mathbb{V}_{P^\star} V_{h+1;\widehat{P}^k}^{\pi^k} \right)(s_h^k, a_h^k) \log(1/\delta)} + \frac{2}{3} \log(1/\delta) \qquad (36)
$$

This directly indicates that

$$P(\mathbb{I}\{\mathcal{E}_2\}) \geq 1 - \delta. \tag{37}$$

Then, we propose the following lemma.

**Lemma 12** (Bound of sum of mean value differences for online RL). *Under event $\mathcal{E}_1$, we have*

$$\sum_{k \in [K-1] \setminus \mathcal{K}} \sum_{h=0}^{H-1} \left| \mathbb{E}_{s' \sim \widehat{P}^k(s_h^k, a_h^k)} V_{h+1; \widehat{P}^k}^{\pi^k}(s') - \mathbb{E}_{s' \sim P^\star(s_h^k, a_h^k)} V_{h+1; \widehat{P}^k}^{\pi^k}(s') \right|$$

$$\lesssim \sqrt{\sum_{k \in [K-1] \setminus \mathcal{K}} \sum_{h=0}^{H-1} \left[ (\mathbb{V}_{P^\star} V_{h+1; \widehat{P}^k}^{\pi^k})(s_h^k, a_h^k) \right] \cdot DE_1(\Psi, \mathcal{S} \times \mathcal{A}, 1/KH) \cdot \log(K |\mathcal{P}| /\delta) \log(KH)}$$

$$+ DE_1(\Psi, \mathcal{S} \times \mathcal{A}, 1/KH) \cdot \log(K |\mathcal{P}| /\delta) \log(KH).$$

*Proof of Lemma 12.* Under event $\mathcal{E}_1$, we have

$$\sum_{k \in [K-1] \setminus \mathcal{K}} \sum_{h=0}^{H-1} \left| \mathbb{E}_{s' \sim \widehat{P}^k(s_h^k, a_h^k)} V_{h+1; \widehat{P}^k}^{\pi^k}(s') - \mathbb{E}_{s' \sim P^\star(s_h^k, a_h^k)} V_{h+1; \widehat{P}^k}^{\pi^k}(s') \right|$$

$$\leq 4 \sum_{k \in [K-1] \setminus \mathcal{K}} \sum_{h=0}^{H-1} \left[ \sqrt{(\mathbb{V}_{P^\star} V_{h+1; \widehat{P}^k}^{\pi^k})(s_h^k, a_h^k) D_\triangle \left( V_{h+1; \widehat{P}^k}^{\pi^k}(s' \sim P^\star(s_h^k, a_h^k)) \parallel V_{h+1; \widehat{P}^k}^{\pi^k}(s' \sim \widehat{P}^k(s_h^k, a_h^k)) \right)} \right]$$

$$+ 5 \sum_{k \in [K-1] \setminus \mathcal{K}} \sum_{h=0}^{H-1} \left[ D_\triangle \left( V_{h+1; \widehat{P}^k}^{\pi^k}(s' \sim P^\star(s_h^k, a_h^k)) \parallel V_{h+1; \widehat{P}^k}^{\pi^k}(s' \sim \widehat{P}^k(s_h^k, a_h^k)) \right) \right]$$

$$\leq 8 \sum_{k \in [K-1] \setminus \mathcal{K}} \sum_{h=0}^{H-1} \left[ \sqrt{(\mathbb{V}_{P^\star} V_{h+1; \widehat{P}^k}^{\pi^k})(s_h^k, a_h^k) \mathbb{H}^2 \left( V_{h+1; \widehat{P}^k}^{\pi^k}(s' \sim P^\star(s_h^k, a_h^k)) \parallel V_{h+1; \widehat{P}^k}^{\pi^k}(s' \sim \widehat{P}^k(s_h^k, a_h^k)) \right)} \right]$$

$$+ 20 \sum_{k \in [K-1] \setminus \mathcal{K}} \sum_{h=0}^{H-1} \left[ \mathbb{H}^2 \left( V_{h+1; \widehat{P}^k}^{\pi^k}(s' \sim P^\star(s_h^k, a_h^k)) \parallel V_{h+1; \widehat{P}^k}^{\pi^k}(s' \sim \widehat{P}^k(s_h^k, a_h^k)) \right) \right]$$

$$\leq 8 \sum_{k \in [K-1] \setminus \mathcal{K}} \sum_{h=0}^{H-1} \left[ \sqrt{(\mathbb{V}_{P^\star} V_{h+1; \widehat{P}^k}^{\pi^k})(s_h^k, a_h^k) \mathbb{H}^2 \left( P^\star(s_h^k, a_h^k) \parallel \widehat{P}^k(s_h^k, a_h^k) \right)} \right]$$

$$+ 20 \sum_{k \in [K-1] \setminus \mathcal{K}} \sum_{h=0}^{H-1} \left[ \mathbb{H}^2 \left( P^\star(s_h^k, a_h^k) \parallel \widehat{P}^k(s_h^k, a_h^k) \right) \right]$$

$$\leq 8 \sqrt{\sum_{k \in [K-1] \setminus \mathcal{K}} \sum_{h=0}^{H-1} \left[ (\mathbb{V}_{P^\star} V_{h+1; \widehat{P}^k}^{\pi^k})(s_h^k, a_h^k) \right] \cdot \sum_{k \in [K-1] \setminus \mathcal{K}} \sum_{h=0}^{H-1} \left[ \mathbb{H}^2 \left( P^\star(s_h^k, a_h^k) \parallel \widehat{P}^k(s_h^k, a_h^k) \right) \right]}$$

$$+ 20 \sum_{k \in [K-1] \setminus \mathcal{K}} \sum_{h=0}^{H-1} \left[ \mathbb{H}^2 \left( P^\star(s_h^k, a_h^k) \parallel \widehat{P}^k(s_h^k, a_h^k) \right) \right]$$

$$\lesssim \sqrt{\sum_{k \in [K-1] \setminus \mathcal{K}} \sum_{h=0}^{H-1} \left[ (\mathbb{V}_{P^\star} V_{h+1; \widehat{P}^k}^{\pi^k})(s_h^k, a_h^k) \right] \cdot DE_1(\Psi, \mathcal{S} \times \mathcal{A}, 1/KH) \cdot \log(K |\mathcal{P}| /\delta) \log(KH)}$$

$$+ DE_1(\Psi, \mathcal{S} \times \mathcal{A}, 1/KH) \cdot \log(K |\mathcal{P}| /\delta) \log(KH), \tag{38}$$

where in the first inequality, we use Lemma 1 to bound the difference of two means $\mathbb{E}_{s' \sim P^\star(s_h^k, a_h^k)} V_{h+1; \widehat{P}^k}^{\pi^k}(s') - \mathbb{E}_{s' \sim \widehat{P}^k(s_h^k, a_h^k)} V_{h+1; \widehat{P}^k}^{\pi^*}(s')$ using variances and the triangle discrimination; in the second inequality we use the fact that that triangle discrimination is equivalent to squared Hellinger distance, i.e., $D_\triangle \leq 4\mathbb{H}^2$; the third inequality is via data processing inequality on the squared Hellinger distance; the fourth inequality is by the Cauchy–Schwarz inequality; the last inequality holds under $\mathcal{E}_1$ by Equation 32. $\square$

The next lemma shows that the event $\mathcal{E}_3$ happens with high probability.

**Lemma 13** (Recursion Event Lemma). *Event $\mathcal{E}_3$ happens with high probability. Specifically, we have*

$$P(\mathbb{I}\{\mathcal{E}_3\}) \geq 1 - (1 + \lceil \log_2(\frac{KH}{G}) \rceil)\delta. \tag{39}$$

*Proof of Lemma 13.* Let $\Delta_{h+1}^{\pi^k} := V_{h+1;\widehat{P}^k}^{\pi^k} - V_{h+1}^{\pi^k}$. First, under event $\mathcal{E}_1$, with happens with probability at least $1 - \delta$ by Equation 31, and also note that $\pi^k$ is deterministic for any $k \in [K-1]$, we can prove the following

$$\sum_{k \in [K-1] \setminus \mathcal{K}} \sum_{h=0}^{H-1} \left[ \left| (\Delta_h^{\pi^k})(s_h^k) - (P^\star \Delta_{h+1}^{\pi^k})(s_h^k, a_h^k) \right| \right] \tag{40}$$

$$= \sum_{k \in [K-1] \setminus \mathcal{K}} \sum_{h=0}^{H-1} \left[ \left| (V_{h;\widehat{P}^k}^{\pi^k})(s_h^k) - (P^\star V_{h+1;\widehat{P}^k}^{\pi^k})(s_h^k, a_h^k) - \left( (V_h^{\pi^k})(s_h^k) - (P^\star V_{h+1}^{\pi^k})(s_h^k, a_h^k) \right) \right| \right]$$

$$= \sum_{k \in [K-1] \setminus \mathcal{K}} \sum_{h=0}^{H-1} \left[ \left| r(s_h^k, a_h^k) + (\widehat{P}^k V_{h+1;\widehat{P}^k}^{\pi^k})(s_h^k, a_h^k) - (P^\star V_{h+1;\widehat{P}^k}^{\pi^k})(s_h^k, a_h^k) - r(s_h^k, a_h^k) \right| \right]$$

$$= \sum_{k \in [K-1] \setminus \mathcal{K}} \sum_{h=0}^{H-1} \left[ \left| (\widehat{P}^k V_{h+1;\widehat{P}^k}^{\pi^k})(s_h^k, a_h^k) - (P^\star V_{h+1;\widehat{P}^k}^{\pi^k})(s_h^k, a_h^k) \right| \right]$$

$$= \sum_{k \in [K-1] \setminus \mathcal{K}} \sum_{h=0}^{H-1} \left[ \left| \mathbb{E}_{s' \sim P^\star(s_h^k, a_h^k)} \left[ V_{h+1;\widehat{P}^k}^{\pi^k}(s') \right] - \mathbb{E}_{s' \sim \widehat{P}^k(s_h^k, a_h^k)} \left[ V_{h+1;\widehat{P}^k}^{\pi^k}(s') \right] \right| \right]$$

$$\lesssim \sqrt{\sum_{k \in [K-1] \setminus \mathcal{K}} \sum_{h=0}^{H-1} \left[ (\mathbb{V}_{P^\star} V_{h+1;\widehat{P}^k}^{\pi^k})(s_h^k, a_h^k) \right] \cdot \mathrm{DE}_1(\Psi, \mathcal{S} \times \mathcal{A}, 1/KH) \cdot \log(K |\mathcal{P}|/\delta) \log(KH)}$$

$$\quad + \mathrm{DE}_1(\Psi, \mathcal{S} \times \mathcal{A}, 1/KH) \cdot \log(K |\mathcal{P}|/\delta) \log(KH)$$

$$= G \tag{41}$$

where the first equality is by the definition of $\Delta_{h+1}^{\pi^k}$, the inequality holds under $\mathcal{E}_1$ by Lemma 12, and the last equality is by definition of $A$ and $G$.

Under event $\mathcal{E}_1$, with probability at least $1 - \lceil \log_2(\frac{KH}{G}) \rceil \delta$, for any $m \in [0, \lceil \log_2(\frac{KH}{G}) \rceil]$

$$C_m = \sum_{k \in [K-1] \setminus \mathcal{K}} \sum_{h=0}^{H-1} \left[ \left( \mathbb{V}_{P^\star} (V_{h+1;\widehat{P}^k}^{\pi^k} - V_{h+1}^{\pi^k})^{2^m} \right)(s_h^k, a_h^k) \right]$$

$$= \sum_{k \in [K-1] \setminus \mathcal{K}} \sum_{h=0}^{H-1} \left[ \left( P^\star (\Delta_{h+1}^{\pi^k})^{2^{m+1}} \right)(s_h^k, a_h^k) - \left( (P^\star (\Delta_{h+1}^{\pi^k})^{2^m})(s_h^k, a_h^k) \right)^2 \right]$$

$$= \sum_{k \in [K-1] \setminus \mathcal{K}} \sum_{h=0}^{H-1} \left[ (\Delta_{h+1}^{\pi^k})^{2^{m+1}}(s_{h+1}^k) \right] - \sum_{k \in [K-1] \setminus \mathcal{K}} \sum_{h=0}^{H-1} \left[ \left( (P^\star (\Delta_{h+1}^{\pi^k})^{2^m})(s_h^k, a_h^k) \right)^2 \right]$$

$$\quad + \sum_{k \in [K-1] \setminus \mathcal{K}} \sum_{h=0}^{H-1} \left( \mathbb{E}_{s \sim P^\star(s_h^k, a_h^k)} \left[ (\Delta_{h+1}^{\pi^k})^{2^{m+1}}(s) \right] - (\Delta_{h+1}^{\pi^k})^{2^{m+1}}(s_{h+1}^k) \right)$$

$$\leq \sum_{k \in [K-1] \setminus \mathcal{K}} \sum_{h=0}^{H-1} \left[ (\Delta_h^{\pi^k})^{2^{m+1}}(s_h^k) - \left( (P^\star (\Delta_{h+1}^{\pi^k})^{2^m})(s_h^k, a_h^k) \right)^2 \right]$$

$$\quad + \sum_{k \in [K-1] \setminus \mathcal{K}} \sum_{h=0}^{H-1} \left( \mathbb{E}_{s \sim P^\star(s_h^k, a_h^k)} \left[ (\Delta_{h+1}^{\pi^k})^{2^{m+1}}(s) \right] - (\Delta_{h+1}^{\pi^k})^{2^{m+1}}(s_h^k) \right)$$

$$\lesssim \sum_{k \in [K-1] \setminus \mathcal{K}} \sum_{h=0}^{H-1} \left[ (\Delta_h^{\pi^k})^{2^{m+1}}(s_h^k) - \left( (P^\star (\Delta_{h+1}^{\pi^k})^{2^m})(s_h^k, a_h^k) \right)^2 \right] + \log(1/\delta)$$

$$\quad + \sqrt{\sum_{k \in [K-1] \setminus \mathcal{K}} \sum_{h=0}^{H-1} \mathbb{V}_{P^\star} \left( (V_{h+1;\widehat{P}^k}^{\pi^k} - V_{h+1}^{\pi^k})^{2^{m+1}} \right)(s_h^k, a_h^k) \log(1/\delta)}$$

$$
\begin{aligned}
&= \sum_{k \in [K-1] \setminus \mathcal{K}} \sum_{h=0}^{H-1} \left[ \left( (\Delta_h^{\pi^k})^{2^m}(s_h^k) + (P^\star (\Delta_{h+1}^{\pi^k})^{2^m})(s_h^k, a_h^k) \right) \cdot \left( (\Delta_h^{\pi^k})^{2^m}(s_h^k) - (P^\star (\Delta_{h+1}^{\pi^k})^{2^m})(s_h^k, a_h^k) \right) \right] \\
&\qquad + \sqrt{\log(1/\delta) \cdot C_{m+1}} + \log(1/\delta) \\
&= \sum_{k \in [K-1] \setminus \mathcal{K}} \sum_{h=0}^{H-1} \left[ \left( (\Delta_h^{\pi^k})^{2^m}(s_h^k) + (P^\star (\Delta_{h+1}^{\pi^k})^{2^m})(s_h^k, a_h^k) \right) \cdot \left( (\Delta_h^{\pi^k})^{2^m}(s_h^k) - (P^\star ((\Delta_{h+1}^{\pi^k})^2)^{2^{m-1}})(s_h^k, a_h^k) \right) \right] \\
&\qquad + \sqrt{\log(1/\delta) \cdot C_{m+1}} + \log(1/\delta) \\
&\leq \sum_{k \in [K-1] \setminus \mathcal{K}} \sum_{h=0}^{H-1} \left[ \left( (\Delta_h^{\pi^k})^{2^m}(s_h^k) + (P^\star (\Delta_{h+1}^{\pi^k})^{2^m})(s_h^k, a_h^k) \right) \cdot \left( (\Delta_h^{\pi^k})^{2^m}(s_h^k) - ((P^\star (\Delta_{h+1}^{\pi^k})^2)(s_h^k, a_h^k))^{2^{m-1}} \right) \right] \\
&\qquad + \sqrt{\log(1/\delta) \cdot C_{m+1}} + \log(1/\delta) \\
&\leq 2^m \sum_{k \in [K-1] \setminus \mathcal{K}} \sum_{h=0}^{H-1} \left[ \left| (\Delta_h^{\pi^k})^2(s_h^k) - ((P^\star \Delta_{h+1}^{\pi^k})(s_h^k, a_h^k))^2 \right| \right] + \sqrt{\log(1/\delta) \cdot C_{m+1}} + \log(1/\delta) \\
&= 2^m \sum_{k \in [K-1] \setminus \mathcal{K}} \sum_{h=0}^{H-1} \left[ \left| \left( (\Delta_h^{\pi^k})(s_h^k) + (P^\star \Delta_{h+1}^{\pi^k})(s_h^k, a_h^k) \right) \cdot \left( (\Delta_h^{\pi^k})(s_h^k) - (P^\star \Delta_{h+1}^{\pi^k})(s_h^k, a_h^k) \right) \right| \right] \\
&\qquad + \sqrt{\log(1/\delta) \cdot C_{m+1}} + \log(1/\delta) \\
&\leq 2 \cdot 2^m \sum_{k \in [K-1] \setminus \mathcal{K}} \sum_{h=0}^{H-1} \left[ \left| (\Delta_h^{\pi^k})(s_h^k) - \left( P^\star \Delta_{h+1}^{\pi^k} \right)(s_h^k, a_h^k) \right| \right] + \sqrt{\log(1/\delta) \cdot C_{m+1}} + \log(1/\delta) \\
&\lesssim 2^m G + \sqrt{\log(1/\delta) \cdot C_{m+1}} + \log(1/\delta) \,, \qquad\qquad\qquad\qquad\qquad\qquad\qquad (42)
\end{aligned}
$$

where in the first inequality we change the index, the second inequality holds with probability at least $1 - \delta$ by Azuma Bernstain's inequality, the third inequality holds because that $E[X^{2^{m-1}}] \geq (E[X])^{2^{m-1}}$ for $m \geq 1$ and $X \geq 0$, the fourth inequality holds by keep using $a^2 - b^2 = (a+b)(a-b)$, then with $E[X^2] \geq E[X]^2$, and the assumption that the trajectory-wise total reward is normalized in $[0, 1]$, the last inequality holds under $\mathcal{E}_1$ by Equation 41, and we take a union bound to get this hold for all $m \in [0, \lceil \log_2(\frac{KH}{G}) \rceil]$ with probability at least $1 - \lceil \log_2(\frac{KH}{G}) \rceil \delta$ (because for each $m \in [0, \lceil \log_2(\frac{KH}{G}) \rceil]$ we need to apply the Azuma Bernstain's inequality once).

The above reasoning directly implies that

$$
P(\mathbb{I}\{\mathcal{E}_3\}) \geq 1 - (1 + \lceil \log_2(\frac{KH}{G}) \rceil) \delta. \qquad\qquad (43)
$$

$\square$

Under the event $\mathcal{E}_3$, we prove the following lemma to bound $\sum_{k \in [K-1] \setminus \mathcal{K}} \sum_{h=0}^{H-1} \left[ (\mathbb{V}_{P^\star} V_{h+1; \widehat{P}^k}^{\pi^k})(s_h^k, a_h^k) \right]$.

**Lemma 14** (Variance Conversion Lemma for online RL). *Under event $\mathcal{E}_3$, we have*

$$
\sum_{k \in [K-1] \setminus \mathcal{K}} \sum_{h=0}^{H-1} \left[ (\mathbb{V}_{P^\star} V_{h+1; \widehat{P}^k}^{\pi^k})(s_h^k, a_h^k) \right]
$$

$$
\leq O\left( \sum_{k \in [K-1] \setminus \mathcal{K}} \sum_{h=0}^{H-1} \left[ (\mathbb{V}_{P^\star} V_{h+1}^{\pi^k})(s_h^k, a_h^k) \right] + DE_1(\Psi, \mathcal{S} \times \mathcal{A}, 1/KH) \cdot \log(K|\mathcal{P}|/\delta) \log(KH) \right).
$$

*Proof of Lemma 14.* Under $\mathcal{E}_3$, we have for any $m \in [0, \lceil \log_2(\frac{KH}{G}) \rceil]$

$$
C_m \lesssim 2^m G + \sqrt{\log(1/\delta) \cdot C_{m+1}} + \log(1/\delta) \,. \qquad\qquad (44)
$$

Then, by Lemma 11, we have

$$
C_0 \lesssim G \,. \qquad\qquad (45)
$$

Also note that we have $A \le 2B + 2C_0$ since $\mathbb{V}_{P^\star}(a+b) \le 2\mathbb{V}_{P^\star}(a) + 2\mathbb{V}_{P^\star}(b)$. Therefore, we have

$$
\begin{aligned}
A &\le 2B + 2C_0 \\
&\lesssim B + G \\
&= B + \sqrt{A \cdot \text{DE}_1(\Psi, \mathcal{S} \times \mathcal{A}, 1/KH) \cdot \log(K\,|\mathcal{P}|/\delta)\log(KH)} \\
&\quad + \text{DE}_1(\Psi, \mathcal{S} \times \mathcal{A}, 1/KH) \cdot \log(K\,|\mathcal{P}|/\delta)\log(KH)
\end{aligned}
\tag{46}
$$

Then, with the fact that $x \le 2a + b^2$ if $x \le a + b\sqrt{x}$, we have

$$
A \le O\!\left( B + \text{DE}_1(\Psi, \mathcal{S} \times \mathcal{A}, 1/KH) \cdot \log(K\,|\mathcal{P}|/\delta)\log(KH) \right),
\tag{47}
$$

which is

$$
\sum_{k\in[K-1]\setminus\mathcal{K}} \sum_{h=0}^{H-1} \left[ \left(\mathbb{V}_{P^\star} V_{h+1;\widehat{P}^k}^{\pi^k}\right)(s_h^k, a_h^k) \right]
$$
$$
\le O\!\left( \sum_{k\in[K-1]\setminus\mathcal{K}} \sum_{h=0}^{H-1} \left[ \left(\mathbb{V}_{P^\star} V_{h+1}^{\pi^k}\right)(s_h^k, a_h^k) \right] + \text{DE}_1(\Psi, \mathcal{S} \times \mathcal{A}, 1/KH) \cdot \log(K\,|\mathcal{P}|/\delta)\log(KH) \right)
\tag{48}
$$

$\square$

By the same reasoning in Lemma 26 of Zhou et al. (2023), we have that with probability at least $1 - \delta$

$$
\sum_{k\in[K-1]\setminus\mathcal{K}} \sum_{h=0}^{H-1} \left[ \left(\mathbb{V}_{P^\star} V_{h+1}^{\pi^k}\right)(s_h^k, a_h^k) \right] \le \sum_{k=0}^{K-1} \sum_{h=0}^{H-1} \left[ \left(\mathbb{V}_{P^\star} V_{h+1}^{\pi^k}\right)(s_h^k, a_h^k) \right]
$$
$$
\le O\!\left( \sum_{k=0}^{K-1} \text{VaR}_{\pi^k} + \log(1/\delta) \right).
\tag{49}
$$

This indicates that

$$
P(\mathbb{I}\{\mathcal{E}_4\}) \ge 1 - \delta.
\tag{50}
$$

We can use the Azuma Bernstain's inequality to get that with probability at least $1 - \delta$:

$$
\sum_{k=0}^{K-1} \sum_{h=1}^{H} r(s_h^k, a_h^k) - \sum_{k=0}^{K-1} V_{0;P^\star}^{\pi^k} \lesssim \sqrt{\sum_{k=0}^{K-1} \text{VaR}_{\pi^k} \log(1/\delta)} + \log(1/\delta).
\tag{51}
$$

This indicates that

$$
P(\mathbb{I}\{\mathcal{E}_5\}) \ge 1 - \delta.
\tag{52}
$$

Then, together with Lemma 13, Equation 31 and Equation 37, we have

$$
P(\mathbb{I}\{\mathcal{E}\}) \ge 1 - \left(5 + \lceil \log_2(\frac{KH}{G}) \rceil\right)\delta \ge 1 - 5KH\delta.
\tag{53}
$$

Finally, under event $\mathcal{E}$, with all the things above (Equation 33, Equation 35, Equation 36, Lemma 12, Lemma 14), we have

$$
\sum_{k=0}^{K-1} V_{0;P^\star}^\star - \sum_{k=0}^{K-1} V_{0;P^\star}^{\pi^k}
$$
$$
= \sum_{k=0}^{K-1} V_{0;P^\star}^\star - \sum_{k=0}^{K-1} \sum_{h=1}^{H} r(s_h^k, a_h^k) + \sum_{k=0}^{K-1} \sum_{h=1}^{H} r(s_h^k, a_h^k) - \sum_{k=0}^{K-1} V_{0;P^\star}^{\pi^k}
$$
$$
\lesssim |\mathcal{K}| + \sum_{k\in[K-1]\setminus\mathcal{K}} \left( V_{0;\widehat{P}^k}^{\pi^k} - \sum_{h=0}^{H-1} r(s_h^k, a_h^k) \right) + \sqrt{\sum_{k=0}^{K-1} \text{VaR}_{\pi^k} \log(1/\delta)} + \log(1/\delta)
$$
$$
\lesssim \log^2(\log(K\,|\mathcal{P}|/\delta)KH) \cdot DE_1(\Psi, \mathcal{S} \times \mathcal{A}, 1/(\log(K\,|\mathcal{P}|/\delta)KH)) + \sum_{k\in[K-1]\setminus\mathcal{K}} \left( V_{0;\widehat{P}^k}^{\pi^k} - \sum_{h=0}^{H-1} r(s_h^k, a_h^k) \right)
$$

$$
+ \sqrt{\sum_{k=0}^{K-1} \mathrm{VaR}_{\pi^k} \log(1/\delta)} + \log(1/\delta)
$$

$$
\lesssim \log^2(\log(K\,|\mathcal{P}|/\delta)KH) \cdot DE_1(\Psi, \mathcal{S} \times \mathcal{A}, 1/(\log(K\,|\mathcal{P}|/\delta)KH)) + \log(1/\delta)
$$

$$
+ \sqrt{\sum_{k \in [K-1] \setminus \mathcal{K}} \sum_{h=0}^{H-1} \left(\mathbb{V}_{P^\star} V_{h+1;\widehat{P}^k}^{\pi^k}\right)(s_h^k, a_h^k) \log(1/\delta)} + \sqrt{\sum_{k=0}^{K-1} \mathrm{VaR}_{\pi^k} \log(1/\delta)}
$$

$$
+ \sum_{k \in [K-1] \setminus \mathcal{K}} \sum_{h=0}^{H-1} \left| \mathbb{E}_{s' \sim \widehat{P}^k(s_1^k, A^k)} V_{1;\widehat{P}^k}^{\pi^k}(s') - \mathbb{E}_{s' \sim P^*(s_1^k, A^k)} V_{1;\widehat{P}^k}^{\pi^k}(s') \right|
$$

$$
\lesssim \log^2(\log(K\,|\mathcal{P}|/\delta)KH) \cdot DE_1(\Psi, \mathcal{S} \times \mathcal{A}, 1/(\log(K\,|\mathcal{P}|/\delta)KH)) + \sqrt{\sum_{k=0}^{K-1} \mathrm{VaR}_{\pi^k} \log(1/\delta)} + \log(1/\delta)
$$

$$
+ \sqrt{\left(\sum_{k \in [K-1] \setminus \mathcal{K}} \sum_{h=0}^{H-1} \left[\left(\mathbb{V}_{P^\star} V_{h+1}^{\pi^k}\right)(s_h^k, a_h^k)\right] + DE_1(\Psi, \mathcal{S} \times \mathcal{A}, 1/KH) \cdot \log(K\,|\mathcal{P}|/\delta) \log(KH)\right) \cdot \log(1/\delta)}
$$

$$
+ \sqrt{\sum_{k \in [K-1] \setminus \mathcal{K}} \sum_{h=0}^{H-1} \left[\left(\mathbb{V}_{P^\star} V_{h+1;\widehat{P}^k}^{\pi^k}\right)(s_h^k, a_h^k)\right] \cdot DE_1(\Psi, \mathcal{S} \times \mathcal{A}, 1/KH) \cdot \log(K\,|\mathcal{P}|/\delta) \log(KH)}
$$

$$
\lesssim \log^2(\log(K\,|\mathcal{P}|/\delta)KH) \cdot DE_1(\Psi, \mathcal{S} \times \mathcal{A}, 1/(\log(K\,|\mathcal{P}|/\delta)KH)) + \sqrt{\sum_{k=0}^{K-1} \mathrm{VaR}_{\pi^k} \log(1/\delta)} + \log(1/\delta)
$$

$$
+ \sqrt{\left(\sum_{k \in [K-1] \setminus \mathcal{K}} \sum_{h=0}^{H-1} \left[\left(\mathbb{V}_{P^\star} V_{h+1}^{\pi^k}\right)(s_h^k, a_h^k)\right] + DE_1(\Psi, \mathcal{S} \times \mathcal{A}, 1/KH) \cdot \log(K\,|\mathcal{P}|/\delta) \log(KH)\right) \cdot \log(1/\delta)}
$$

$$
+ \sqrt{\left(\sum_{k \in [K-1] \setminus \mathcal{K}} \sum_{h=0}^{H-1} \left[\left(\mathbb{V}_{P^\star} V_{h+1}^{\pi^k}\right)(s_h^k, a_h^k)\right] + DE_1(\Psi, \mathcal{S} \times \mathcal{A}, 1/KH) \cdot \log(K\,|\mathcal{P}|/\delta) \log(KH)\right)}
$$

$$
\times \sqrt{DE_1(\Psi, \mathcal{S} \times \mathcal{A}, 1/KH) \cdot \log(K\,|\mathcal{P}|/\delta) \log(KH))}
$$

$$
\lesssim \log^2(\log(K\,|\mathcal{P}|/\delta)KH) \cdot DE_1(\Psi, \mathcal{S} \times \mathcal{A}, 1/(\log(K\,|\mathcal{P}|/\delta)KH)) + \sqrt{\sum_{k=0}^{K-1} \mathrm{VaR}_{\pi^k} \log(1/\delta)} + \log(1/\delta)
$$

$$
+ \sqrt{\left(\sum_{k=0}^{K-1} \mathrm{VaR}_{\pi^k} + \log(1/\delta) + DE_1(\Psi, \mathcal{S} \times \mathcal{A}, 1/KH) \cdot \log(K\,|\mathcal{P}|/\delta) \log(KH)\right) \cdot \log(1/\delta)}
$$

$$
+ \sqrt{\left(\sum_{k=0}^{K-1} \mathrm{VaR}_{\pi^k} + \log\left(\frac{1}{\delta}\right) + DE_1\left(\Psi, \mathcal{S} \times \mathcal{A}, \frac{1}{KH}\right) \cdot \log\left(\frac{K\,|\mathcal{P}|}{\delta}\right) \log(KH)\right)}
$$

$$
\times \sqrt{DE_1\left(\Psi, \mathcal{S} \times \mathcal{A}, \frac{1}{KH}\right) \log\left(\frac{K\,|\mathcal{P}|}{\delta}\right) \log(KH)}
$$

$$
\leq O\Big(\sqrt{\sum_{k=0}^{K-1} \mathrm{VaR}_{\pi^k} \cdot DE_1(\Psi, \mathcal{S} \times \mathcal{A}, 1/KH) \cdot \log(K\,|\mathcal{P}|/\delta) \log(KH)}
$$

$$
+ DE_1(\Psi, \mathcal{S} \times \mathcal{A}, 1/KH) \cdot \log(K\,|\mathcal{P}|/\delta) \log(KH)\Big). \tag{54}
$$

The final result follows by replacing $\delta$ to be $\delta/(5KH)$ to make the event $\mathcal{E}$ happen with probability at least $1 - \delta$.

## D.2 Proof of Corollary 2

*Proof of Corollary 2.* By Lemma 8, we have

$$\text{VaR}_{\pi^k} = \sum_{h=0}^{H-1} \mathbb{E}_{s,a \sim d_h^{\pi^k}} \left[ \left( \mathbb{V}_{P^\star} V_{h+1}^{\pi^k} \right)(s,a) \right] \tag{55}$$

Therefore, when $P^\star$ is deterministic, the $\mathbb{E}_{s,a \sim d_h^{\pi^k}} \left[ \left( \mathbb{V}_{P^\star} V_{h+1}^{\pi^k} \right)(s,a) \right]$ terms are all 0 for any $k \in [K-1]$ and $h \in [H-1]$, and then the $\sum_{k=0}^{K-1} \text{VaR}_{\pi^k}$ term in the higher order term in Theorem 1 is 0. $\qquad\square$

## D.3 Proof of Corollary 3

*Proof of Corollary 3.* We follow the MLE guarantee for the infinite model class in Lemma 10 and the same proof steps in the proof of Theorem 1 in Appendix D.1. $\qquad\square$

# E Detailed Proofs for the Offline RL setting in Section 5

## E.1 Proof of Theorem 2

The following is the full proof of Theorem 2.

*Proof of Theorem 2.* First, by the realizability assumption, the standard generalization bound for MLE (Lemma 9) with simply setting $D_i$ to be the delta distribution on the $(s_h^k, a_h^k)$ pairs in the offline dataset $\mathcal{D}$, we have that w.p. at least $1 - \delta$:

(1) $P^\star \in \widehat{\mathcal{P}}$;

(2)
$$\frac{1}{K} \sum_{k=1}^{K} \sum_{h=0}^{H-1} \mathbb{H}^2(P^\star(s_h^k, a_h^k) \| \widehat{P}(s_h^k, a_h^k)) \leq \frac{22 \log(|\mathcal{P}|/\delta)}{K}. \tag{56}$$

Then, with the above realizability in (1), and by the pessimistic algorithm design $\hat{\pi} \leftarrow \text{argmax}_{\pi \in \widehat{\mathcal{P}}} \min_{P \in \widehat{\mathcal{P}}} V_{0;P}^\pi(s_0)$, $\widehat{P} \leftarrow \text{argmin}_{P \in \widehat{\mathcal{P}}} V_{0;P}^{\hat{\pi}}(s_0)$, we have that for any $\pi^\star \in \Pi$

$$
\begin{aligned}
V_{0;P^\star}^{\pi^\star} - V_{0;P^\star}^{\hat{\pi}} &= V_{0;P^\star}^{\pi^\star} - V_{0;\widehat{P}}^{\pi^\star} + V_{0;\widehat{P}}^{\pi^\star} - V_{0;P^\star}^{\hat{\pi}} \\
&\leq V_{0;P^\star}^{\pi^\star} - V_{0;\widehat{P}}^{\pi^\star} + V_{0;\widehat{P}}^{\hat{\pi}} - V_{0;P^\star}^{\hat{\pi}} \\
&\leq V_{0;P^\star}^{\pi^\star} - V_{0;\widehat{P}}^{\pi^\star}.
\end{aligned}
\tag{57}
$$

We can then bound $V_{0;P^\star}^{\pi^\star} - V_{0;\widehat{P}}^{\pi^\star}$ using the simulation lemma (Lemma 7):

$$V_{0;P^\star}^{\pi^\star} - V_{0;\widehat{P}}^{\pi^\star} \leq \sum_{h=0}^{H-1} \mathbb{E}_{s,a \sim d_h^{\pi^\star}} \left[ \left| \mathbb{E}_{s' \sim P^\star(s,a)} V_{h+1;\widehat{P}}^{\pi^\star}(s') - \mathbb{E}_{s' \sim \widehat{P}(s,a)} V_{h+1;\widehat{P}}^{\pi^\star}(s') \right| \right]. \tag{58}$$

Then, we prove the following lemma to bound the RHS of Equation 58.

**Lemma 15** (Bound of sum of mean value differences for offline RL)**.** *With probability at least $1 - \delta$, we have*

$$\sum_{h=0}^{H-1} \mathbb{E}_{s,a \sim d_h^{\pi^\star}} \left[ \left| \mathbb{E}_{s' \sim P^\star(s,a)} V_{h+1;\widehat{P}}^{\pi^\star}(s') - \mathbb{E}_{s' \sim \widehat{P}(s,a)} V_{h+1;\widehat{P}}^{\pi^\star}(s') \right| \right]$$

$$\leq 8 \sqrt{ \sum_{h=0}^{H-1} \mathbb{E}_{s,a \sim d_h^{\pi^\star}} \left[ \left( \mathbb{V}_{P^\star} V_{h+1;\widehat{P}}^{\pi^\star} \right)(s,a) \right] \cdot \frac{22 C^{\pi^\star} \log(|\mathcal{P}|/\delta)}{K} } + \frac{440 C^{\pi^\star} \log(|\mathcal{P}|/\delta)}{K}.$$

*Proof of Lemma 15.* We have

$$\sum_{h=0}^{H-1} \mathbb{E}_{s,a \sim d_h^{\pi^\star}} \left[ \left| \mathbb{E}_{s' \sim P^\star(s,a)} V_{h+1;\widehat{P}}^{\pi^\star}(s') - \mathbb{E}_{s' \sim \widehat{P}(s,a)} V_{h+1;\widehat{P}}^{\pi^\star}(s') \right| \right]$$

$$\leq 4 \sum_{h=0}^{H-1} \mathbb{E}_{s,a \sim d_h^{\pi^\star}} \left[ \sqrt{(\mathbb{V}_{P^\star} V_{h+1;\widehat{P}}^{\pi^\star})(s,a) D_\triangle \left( V_{h+1;\widehat{P}}^{\pi^\star}(s' \sim P^\star(s,a)) \parallel V_{h+1;\widehat{P}}^{\pi^\star}(s' \sim \widehat{P}(s,a)) \right)} \right]$$

$$+ 5 \sum_{h=0}^{H-1} \mathbb{E}_{s,a \sim d_h^{\pi^\star}} \left[ D_\triangle \left( V_{h+1;\widehat{P}}^{\pi^\star}(s' \sim P^\star(s,a)) \parallel V_{h+1;\widehat{P}}^{\pi^\star}(s' \sim \widehat{P}(s,a)) \right) \right]$$

$$\leq 8 \sum_{h=0}^{H-1} \mathbb{E}_{s,a \sim d_h^{\pi^\star}} \left[ \sqrt{(\mathbb{V}_{P^\star} V_{h+1;\widehat{P}}^{\pi^\star})(s,a) \mathbb{H}^2 \left( V_{h+1;\widehat{P}}^{\pi^\star}(s' \sim P^\star(s,a)) \parallel V_{h+1;\widehat{P}}^{\pi^\star}(s' \sim \widehat{P}(s,a)) \right)} \right]$$

$$+ 20 \sum_{h=0}^{H-1} \mathbb{E}_{s,a \sim d_h^{\pi^\star}} \left[ \mathbb{H}^2 \left( V_{h+1;\widehat{P}}^{\pi^\star}(s' \sim P^\star(s,a)) \parallel V_{h+1;\widehat{P}}^{\pi^\star}(s' \sim \widehat{P}(s,a)) \right) \right]$$

$$\leq 8 \sum_{h=0}^{H-1} \mathbb{E}_{s,a \sim d_h^{\pi^\star}} \left[ \sqrt{(\mathbb{V}_{P^\star} V_{h+1;\widehat{P}}^{\pi^\star})(s,a) \mathbb{H}^2 \left( P^\star(s,a) \parallel \widehat{P}(s,a) \right)} \right] + 20 \sum_{h=0}^{H-1} \mathbb{E}_{s,a \sim d_h^{\pi^\star}} \left[ \mathbb{H}^2 \left( P^\star(s,a) \parallel \widehat{P}(s,a) \right) \right] \tag{59}$$

where in the first inequality, we use Lemma 1 to bound the difference of two means $\mathbb{E}_{s' \sim P^\star(s,a)} V_{h+1;\widehat{P}}^{\pi^\star}(s') - \mathbb{E}_{s' \sim \widehat{P}(s,a)} V_{h+1;\widehat{P}}^{\pi^\star}(s')$ using variances and the triangle discrimination; in the second inequality we use the fact that that triangle discrimination is equivalent to squared Hellinger distance, i.e., $D_\triangle \leq 4\mathbb{H}^2$; the third inequality is via data processing inequality on the squared Hellinger distance. Next, starting from Equation 59, with probability at least $1 - \delta$, we have

$$8 \sum_{h=0}^{H-1} \mathbb{E}_{s,a \sim d_h^{\pi^\star}} \left[ \sqrt{(\mathbb{V}_{P^\star} V_{h+1;\widehat{P}}^{\pi^\star})(s,a) \mathbb{H}^2 \left( P^\star(s,a) \parallel \widehat{P}(s,a) \right)} \right] + 20 \sum_{h=0}^{H-1} \mathbb{E}_{s,a \sim d_h^{\pi^\star}} \left[ \mathbb{H}^2 \left( P^\star(s,a) \parallel \widehat{P}(s,a) \right) \right]$$

$$\leq 8 \sqrt{\sum_{h=0}^{H-1} \mathbb{E}_{s,a \sim d_h^{\pi^\star}} \left[ (\mathbb{V}_{P^\star} V_{h+1;\widehat{P}}^{\pi^\star})(s,a) \right] \cdot \sum_{h=0}^{H-1} \mathbb{E}_{s,a \sim d_h^{\pi^\star}} \left[ \mathbb{H}^2 \left( P^\star(s,a) \parallel \widehat{P}(s,a) \right) \right]}$$

$$+ 20 \sum_{h=0}^{H-1} \mathbb{E}_{s,a \sim d_h^{\pi^\star}} \left[ \mathbb{H}^2 \left( P^\star(s,a) \parallel \widehat{P}(s,a) \right) \right]$$

$$\leq 8 \sqrt{\sum_{h=0}^{H-1} \mathbb{E}_{s,a \sim d_h^{\pi^\star}} \left[ (\mathbb{V}_{P^\star} V_{h+1;\widehat{P}}^{\pi^\star})(s,a) \right] \cdot C^{\pi^\star} \frac{1}{K} \sum_{k=1}^{K} \sum_{h=0}^{H-1} \mathbb{H}^2(P^\star(s_h^k, a_h^k) \| \widehat{P}(s_h^k, a_h^k))}$$

$$+ 20 C^{\pi^\star} \frac{1}{K} \sum_{k=1}^{K} \sum_{h=0}^{H-1} \mathbb{H}^2(P^\star(s_h^k, a_h^k) \| \widehat{P}(s_h^k, a_h^k))$$

$$\leq 8 \sqrt{\sum_{h=0}^{H-1} \mathbb{E}_{s,a \sim d_h^{\pi^\star}} \left[ (\mathbb{V}_{P^\star} V_{h+1;\widehat{P}}^{\pi^\star})(s,a) \right] \cdot \frac{22 C^{\pi^\star} \log(|\mathcal{P}|/\delta)}{K}} + \frac{440 C^{\pi^\star} \log(|\mathcal{P}|/\delta)}{K}, \tag{60}$$

where the first inequality is by the Cauchy–Schwarz inequality; the second inequality is by the definition of single policy coverage (Definition 3); the last inequality holds with probability at least $1 - \delta$ with Equation 56. Substituting Equation 60 into Equation 59 ends our proof. $\qquad\square$

We denote $\mathcal{E}$ as the event that Lemma 15 holds. Under the event $\mathcal{E}$, we prove the following lemma to bound $\sum_{h=0}^{H-1} \mathbb{E}_{s,a \sim d_h^{\pi^\star}} \left[ (\mathbb{V}_{P^\star} V_{h+1;\widehat{P}}^{\pi^\star})(s,a) \right]$ with $\widetilde{O}(\sum_{h=0}^{H-1} \mathbb{E}_{s,a \sim d_h^{\pi^\star}} \left[ (\mathbb{V}_{P^\star} V_{h+1}^{\pi^\star})(s,a) \right] + C^{\pi^\star} \log(|\mathcal{P}|/\delta)/K)$.

**Lemma 16** (Variance Conversion Lemma for offline RL). *Under event $\mathcal{E}$, we have*

$$\sum_{h=0}^{H-1} \mathbb{E}_{s,a \sim d_h^{\pi^\star}} \left[ (\mathbb{V}_{P^\star} V_{h+1;\widehat{P}}^{\pi^\star})(s,a) \right] \leq O\left( \sum_{h=0}^{H-1} \mathbb{E}_{s,a \sim d_h^{\pi^\star}} \left[ (\mathbb{V}_{P^\star} V_{h+1}^{\pi^\star})(s,a) \right] + C^{\pi^\star} \frac{\log(|\mathcal{P}|/\delta)}{K} \right).$$

*Proof of Lemma 16.* For notational simplicity, we denote $A := \sum_{h=0}^{H-1} \mathbb{E}_{s,a \sim d_h^{\pi^\star}} \left[ \left( \mathbb{V}_{P^\star} V_{h+1;\widehat{P}}^{\pi^\star} \right)(s,a) \right]$, and we denote
$B := \sum_{h=0}^{H-1} \mathbb{E}_{s,a \sim d_h^{\pi^\star}} \left[ \left( \mathbb{V}_{P^\star} V_{h+1}^{\pi^\star} \right)(s,a) \right]$, $C := \sum_{h=0}^{H-1} \mathbb{E}_{s,a \sim d_h^{\pi^\star}} \left[ \left( \mathbb{V}_{P^\star} (V_{h+1;\widehat{P}}^{\pi^\star} - V_{h+1}^{\pi^\star}) \right)(s,a) \right]$, then we have

$$A \leq 2B + 2C,$$

since $\mathbb{V}_{P^\star}(a+b) \leq 2\mathbb{V}_{P^\star}(a) + 2\mathbb{V}_{P^\star}(b)$.

Let $\Delta_{h+1}^{\pi^\star} := V_{h+1;\widehat{P}}^{\pi^\star} - V_{h+1}^{\pi^\star}$. Then, w.p. at least $1-\delta$, we have

$$
\begin{aligned}
C &= \sum_{h=0}^{H-1} \mathbb{E}_{s,a \sim d_h^{\pi^\star}} \left[ \left( P^\star (\Delta_{h+1}^{\pi^\star})^2 \right)(s,a) - \left( P^\star \Delta_{h+1}^{\pi^\star} \right)^2 (s,a) \right] \\
&= \sum_{h=0}^{H-1} \mathbb{E}_{s \sim d_{h+1}^{\pi^\star}} \left[ (\Delta_{h+1}^{\pi^\star})^2 (s) \right] - \sum_{h=0}^{H-1} \mathbb{E}_{s,a \sim d_h^{\pi^\star}} \left[ \left( P^\star \Delta_{h+1}^{\pi^\star} \right)^2 (s,a) \right] \\
&\leq \sum_{h=0}^{H-1} \mathbb{E}_{s,a \sim d_h^{\pi^\star}} \left[ (\Delta_h^{\pi^\star})^2 (s) - \left( P^\star \Delta_{h+1}^{\pi^\star} \right)^2 (s,a) \right] \\
&= \sum_{h=0}^{H-1} \mathbb{E}_{s,a \sim d_h^{\pi^\star}} \left[ \left( (\Delta_h^{\pi^\star})(s) + (P^\star \Delta_{h+1}^{\pi^\star})(s,a) \right) \cdot \left( (\Delta_h^{\pi^\star})(s) - (P^\star \Delta_{h+1}^{\pi^\star})(s,a) \right) \right], \quad (61)
\end{aligned}
$$

where the first equality is by the definition of variance, the second equality holds as $d_h^{\pi^\star}$ is the occupancy measure also generated under $P^\star$, the first inequality is just changing the index, the third equality holds as $a^2 - b^2 = (a+b) \cdot (a-b)$. Starting from Equation 61, we have

$$
\begin{aligned}
&\sum_{h=0}^{H-1} \mathbb{E}_{s,a \sim d_h^{\pi^\star}} \left[ \left( (\Delta_h^{\pi^\star})(s) + (P^\star \Delta_{h+1}^{\pi^\star})(s,a) \right) \cdot \left( (\Delta_h^{\pi^\star})(s) - (P^\star \Delta_{h+1}^{\pi^\star})(s,a) \right) \right] \\
&\leq 2 \sum_{h=0}^{H-1} \mathbb{E}_{s,a \sim d_h^{\pi^\star}} \left[ \left| (\Delta_h^{\pi^\star})(s) - (P^\star \Delta_{h+1}^{\pi^\star})(s,a) \right| \right] \\
&= 2 \sum_{h=0}^{H-1} \mathbb{E}_{s,a \sim d_h^{\pi^\star}} \left[ \left| (V_{h;\widehat{P}}^{\pi^\star})(s) - (P^\star V_{h+1;\widehat{P}}^{\pi^\star})(s,a) - \left( (V_h^{\pi^\star})(s) - (P^\star V_{h+1}^{\pi^\star})(s,a) \right) \right| \right] \\
&= 2 \sum_{h=0}^{H-1} \mathbb{E}_{s,a \sim d_h^{\pi^\star}} \left[ \left| r(s,a) + (\widehat{P} V_{h+1;\widehat{P}}^{\pi^\star})(s,a) - (P^\star V_{h+1;\widehat{P}}^{\pi^\star})(s,a) - r(s,a) \right| \right] \\
&= 2 \sum_{h=0}^{H-1} \mathbb{E}_{s,a \sim d_h^{\pi^\star}} \left[ \left| (\widehat{P} V_{h+1;\widehat{P}}^{\pi^\star})(s,a) - (P^\star V_{h+1;\widehat{P}}^{\pi^\star})(s,a) \right| \right], \quad (62)
\end{aligned}
$$

where the inequality holds as the value functions are all bounded by 1 by the assumption that the total reward over any trajectory is bounded by 1, the first equality is by the definition of $\Delta_{h+1}^{\pi^\star}$, the second equality is because $a$ is drawn from $\pi^\star$. Starting from Equation 62, we have

$$
\begin{aligned}
&2 \sum_{h=0}^{H-1} \mathbb{E}_{s,a \sim d_h^{\pi^\star}} \left[ \left| (\widehat{P} V_{h+1;\widehat{P}}^{\pi^\star})(s,a) - (P^\star V_{h+1;\widehat{P}}^{\pi^\star})(s,a) \right| \right] \\
&= 2 \sum_{h=0}^{H-1} \mathbb{E}_{s,a \sim d_h^{\pi^\star}} \left[ \left| \mathbb{E}_{s' \sim P^\star(s,a)} \left[ V_{h+1;\widehat{P}}^{\pi^\star}(s') \right] - \mathbb{E}_{s' \sim \widehat{P}(\cdot|s,a)} \left[ V_{h+1;\widehat{P}}^{\pi^\star}(s') \right] \right| \right] \\
&\leq 16 \sqrt{\sum_{h=0}^{H-1} \mathbb{E}_{s,a \sim d_h^{\pi^\star}} \left[ \left( \mathbb{V}_{P^\star} V_{h+1;\widehat{P}}^{\pi^\star} \right)(s,a) \right] \cdot \frac{22 C^{\pi^\star} \log(|\mathcal{P}|/\delta)}{K}} + \frac{880 C^{\pi^\star} \log(|\mathcal{P}|/\delta)}{K} \\
&= 16 \sqrt{A \cdot \frac{22 C^{\pi^\star} \log(|\mathcal{P}|/\delta)}{K}} + \frac{880 C^{\pi^\star} \log(|\mathcal{P}|/\delta)}{K} \quad (63)
\end{aligned}
$$

where the inequality holds with probability at least $1-\delta$ by Lemma 15, and the second equality is by definition of $A$.

Then combining Equation 61, Equation 62 and Equation 63, we obtain an upper bound for $C$, which suggests

$$A \leq 2B + 2C$$

$$\leq 2B + \frac{1760 C^{\pi^\star} \log(|\mathcal{P}|/\delta)}{K} + 32\sqrt{\frac{22 C^{\pi^\star} \log(|\mathcal{P}|/\delta)}{K}} \cdot \sqrt{A}.$$

Then, with the fact that $x \leq 2a + b^2$ if $x \leq a + b\sqrt{x}$, we have

$$A \leq 4B + \frac{3520 C^{\pi^\star} \log(|\mathcal{P}|/\delta)}{K} + \frac{22528 C^{\pi^\star} \log(|\mathcal{P}|/\delta)}{K} \leq O\big(B + \frac{C^{\pi^\star} \log(|\mathcal{P}|/\delta)}{K}\big).$$

$\square$

With the above lemmas, we can now prove the final results of Theorem 2. We have that w.p. at least $1 - \delta$

$$V_{0;P^\star}^{\pi^\star} - V_{0;P^\star}^{\hat{\pi}} \leq O\Big(\sqrt{A \cdot \frac{C^{\pi^\star} \log(|\mathcal{P}|/\delta)}{K}} + \frac{C^{\pi^\star} \log(|\mathcal{P}|/\delta)}{K}\Big)$$

$$\leq O\Big(\sqrt{\big(B + \frac{C^{\pi^\star} \log(|\mathcal{P}|/\delta)}{K}\big) \cdot \frac{C^{\pi^\star} \log(|\mathcal{P}|/\delta)}{K}} + \frac{C^{\pi^\star} \log(|\mathcal{P}|/\delta)}{K}\Big)$$

$$\leq O\Big(\sqrt{B \cdot \frac{C^{\pi^\star} \log(|\mathcal{P}|/\delta)}{K}} + \sqrt{\frac{C^{\pi^\star} \log(|\mathcal{P}|/\delta)}{K} \cdot \frac{C^{\pi^\star} \log(|\mathcal{P}|/\delta)}{K}} + \frac{C^{\pi^\star} \log(|\mathcal{P}|/\delta)}{K}\Big)$$

$$= O\Big(\sqrt{\sum_{h=0}^{H-1} \mathbb{E}_{s,a\sim d_h^{\pi^\star}} \big[\big(\mathbb{V}_{P^\star} V_{h+1}^{\pi^\star}\big)(s,a)\big] \cdot \frac{C^{\pi^\star} \log(|\mathcal{P}|/\delta)}{K}} + \frac{C^{\pi^\star} \log(|\mathcal{P}|/\delta)}{K}\Big)$$

(64)

$$= O\Big(\sqrt{\frac{\mathrm{VaR}_{\pi^\star} C^{\pi^\star} \log(|\mathcal{P}|/\delta)}{K}} + \frac{C^{\pi^\star} \log(|\mathcal{P}|/\delta)}{K}\Big),$$

where in the last equation we use Lemma 8, and $\mathrm{VaR}_{\pi^\star} := \mathbb{E}\Big[\big(\sum_{h=0}^{H-1} r(s_h, \pi^\star(s_h)) - V_0^{\pi^\star}\big)^2\Big]$. $\square$

## E.2 PROOF OF COROLLARY 4

*Proof of Corollary 4.* By Lemma 8, we have

$$\mathrm{VaR}_{\pi^\star} = \sum_{h=0}^{H-1} \mathbb{E}_{s,a\sim d_h^{\pi^\star}} \big[\big(\mathbb{V}_{P^\star} V_{h+1}^{\pi^\star}\big)(s,a)\big] \tag{65}$$

Therefore, when $P^\star$ is deterministic, the $\mathbb{E}_{s,a\sim d_h^{\pi^\star}}\big[\big(\mathbb{V}_{P^\star} V_{h+1}^{\pi^\star}\big)(s,a)\big]$ terms are all 0 for any $k \in [K-1]$ and $h \in [H-1]$, and then the $\mathrm{VaR}_{\pi^\star}$ term in the higher order term in Theorem 2 is 0. $\square$

## E.3 PROOF OF COROLLARY 5

*Proof of Corollary 5.* This claim follows the proof of Theorem 2, while we take a different choice of $\beta$ that depends on the bracketing number and follow the MLE guarantee in Lemma 10 for infinite model class. $\square$

## E.4 PROOF OF THE CLAIM IN EXAMPLE 2

*Proof.* Recall that in Definition 3, we have

$$C_{\mathcal{D}}^{\pi^*} := \max_{h, P\in\mathcal{P}} \frac{\mathbb{E}_{s,a\sim d_h^{\pi^*}} \mathbb{H}^2\big(P(s,a) \parallel P^\star(s,a)\big)}{1/K \sum_{k=1}^{K} \mathbb{H}^2\big(P(s_h^k, a_h^k) \parallel P^\star(s_h^k, a_h^k)\big)}.$$

For each step $h$, define two distributions, $p_h, q_h$, where $p_h(s,a) = d^{\pi^*}(s,a)$, $q_h(s,a) = \frac{1}{K}\sum_{k=1}^{K} \mathbb{I}\{(s,a) = (s_h^k, a_h^k)\}$, and we define $f(s,a,P) = \mathbb{H}^2(P(s,a) \parallel P^\star(s,a))$, then we have

$$C_{\mathcal{D}}^{\pi^*} = \max_{h, P\in\mathcal{P}} \frac{\mathbb{E}_{s,a\sim p_h} f(s,a,P)}{\mathbb{E}_{s,a\sim q_h} f(s,a,P)}$$

$$= \max_{h, P \in \mathcal{P}} \frac{\mathbb{E}_{s,a \sim q_h} \frac{p_h(s,a)}{q_h(s,a)} f(s,a,P)}{\mathbb{E}_{s,a \sim q_h} f(s,a,P)}$$

$$\leq \max_{h,s,a} \frac{p_h(s,a)}{q_h(s,a)}$$

$$\leq \max_{h,s,a} \frac{1}{q_h(s,a)} . \tag{66}$$

Note that for all $h$, $\{(s_h^k, a_h^k)\}_{k=1}^K$ are i.i.d. samples drawn from $d_h^{\pi^b}$, therefore, $\mathbb{E}[\mathbb{I}\{(s_h^k, a_h^k) = (s,a)\}] = d_h^{\pi^b}(s,a)$. By Hoeffding's inequality and with a union bound over $s, a, h$, and for $K \geq \frac{2\log(\frac{|\mathcal{S}||\mathcal{A}|H}{\delta})}{\rho_{\min}^2}$, w.p. at least $1 - \delta$, we have

$$q_h(s,a) = \frac{1}{K} \sum_{k=1}^K \mathbb{I}\{(s_h^k, a_h^k) = (s,a)\}$$

$$\geq d_h^{\pi^b}(s,a) - \sqrt{\frac{\log(\frac{|\mathcal{S}||\mathcal{A}|H}{\delta})}{2K}}$$

$$\geq \frac{d_h^{\pi^b}(s,a)}{2} , \tag{67}$$

where in the last inequality we use the assumption that $d_h^{\pi^b}(s,a) \geq \rho_{\min}, \forall s,a,h$, which gives us $K \geq \frac{2\log(\frac{|\mathcal{S}||\mathcal{A}|H}{\delta})}{\rho_{\min}^2} \geq \max_{s,a,h} \frac{2\log(\frac{|\mathcal{S}||\mathcal{A}|H}{\delta})}{(d_h^{\pi^b}(s,a))^2}$, so $K \geq \frac{2\log(\frac{|\mathcal{S}||\mathcal{A}|H}{\delta})}{(d_h^{\pi^b}(s,a))^2}$ for any $s,a,h$.

Therefore, with $K \geq \frac{2\log(\frac{|\mathcal{S}||\mathcal{A}|H}{\delta})}{\rho_{\min}^2}$, we have that w.p. at least $1 - \delta$

$$C_{\mathcal{D}}^{\pi^*} \leq \max_{h,s,a} \frac{1}{q_h(s,a)} \leq \max_{h,s,a} \frac{2}{d_h^{\pi^b}(s,a)} \leq \frac{2}{\rho_{min}} . \tag{68}$$

$\square$