# OpenReview forum: "Model-based RL as a Minimalist Approach to Horizon-Free and Second-Order Bounds"
_ICLR.cc/2025/Conference — ICLR 2025 Poster_

### Official Review · Reviewer_XvPg · 2024-11-03

**Soundness:** 3
**Presentation:** 2
**Contribution:** 3
**Rating:** 6
**Confidence:** 3

**Summary:**

The paper studies model-based RL in MDPs with possibly infinite state-action spaces and total returns bounded in [0,1]. The authors show that a simple approach consisting of learning a version space around the MLE of the transition kernel and performing optimistic/pessimistic planning on top of it achieves strong horizon-free and second-order bounds in online/offline RL.

**Strengths:**

1. The paper presents novel and strong results. To my knowledge, the ones studied here are the simplest algorithms to be shown to achieve both horizon-free and second-order bounds (eg, most prior works required specific algorithmic tweaks to this purpose). Due to the popularity of MLE-based approaches, I think this is a relevant result
2. The paper studies both the online and offline RL settings
3. I particularly appreciated the corollaries for MDPs with deterministic transition kernel, which I think is an understudied setting despite its practical relevance
4. Although I only skimmed through the proofs, they do seem sound
5. Relevant literature is properly discussed

**Weaknesses:**

1. My main concern is that the paper is quite hard to follow, extremely technical, and lacks some clarity in many points. More precisely, the paper is basically a list of theorems/lemmas/corollaries/definitions without too many explanations: several terms that may only be familiar to experts in this line of works are not introduced with sufficient detail and many choices are not motivated. For example,
- it is not clear why the assumption that trajectory returns are in [0,1] is reasonable in practice
- for offline RL, it is not clear why the assumption that the dataset contains trajectories is needed (lines 181-191). Can't you have only transitions?
- the definition of Eluder dimension is not clear (eg what's an epsilon-independent sequence?). It is clearer in prior works
- not clear why the bounds use the function class \Psi defined at lines 258-259. Eg, why not using P itself, which after all is the hypothesis space used for learning?
- In Section 4, it is never explicitly stated that the results before the paragraph "Extension to infinite class P" focus on finite P (the reader has to figure it out from the definition of beta)
- not clear why using bracketing numbers in the analysis. Eg how do they compare to the more standard covering numbers (if there is any difference)? Also, it be great to give examples of what those are for "common" function classes (eg when P is linear)
- not clear how the "single-policy coverage" of Def. 3 compares to the more standard concentrability coefficients defined as ratios between policy visitation distributions used in the offline RL literature
2. The algorithms studied here are both intractable due to the fact that they have to build a version space of transition kernels around the MLE. This is only tractable for simple function classes (eg finite) and limits the practical relevance of this work. Also, I feel this breaks a bit the overall story about the "simplicity" of the approaches. Eg, the conclusions state "we presented a minimalist approach for achieving nearly horizon-free and second-order bounds in online and offline RL: simply train transition models via Maximum Likelihood Estimation followed by optimistic or pessimistic planning", but I don't think this is properly correct: it gives the impression that all we have to do is MLE, but we do have to build version spaces around it, and that's not an easy thing.
3. Lines 171-172: "Without loss of generality, we assume V \in [0,1]". I don't think this is without loss of generality, as the filtering approach proposed in footnote 2 would require doing policy-evaluation for many policies and transition kernels, which is what we want to avoid during training in the first place with an MLE approach
4. [minor] the abstract is very verbose, and I felt many details could have been omitted without altering its quality

**Questions:**

1. See clarification questions above
2. The bound in Example 1 for tabular MDPs seems to have a worse dependence in SA that existing bounds for the same setting (see eg Jin et al. 2020 or Appendix C in Al-Marjani et al. 2023) despite having a better dependence in H. Any intuition why this happens?

[Jin et al. 2020] Jin, C., Krishnamurthy, A., Simchowitz, M., & Yu, T. (2020, November). Reward-free exploration for reinforcement learning. In International Conference on Machine Learning (pp. 4870-4879). PMLR.
[Al-Marjani et al. 2023] Al-Marjani, A., Tirinzoni, A., & Kaufmann, E. (2023, July). Active coverage for pac reinforcement learning. In The Thirty Sixth Annual Conference on Learning Theory (pp. 5044-5109). PMLR.

---

> ### Author Response · Authors · 2024-11-21
>
> We sincerely thank you for your constructive feedback and valuable suggestions. Below, we address your concerns in detail:
>
> **Q1**: My main concern is that the paper is quite hard to follow, extremely technical, and lacks some clarity in many points. More precisely, the paper is basically a list of theorems/lemmas/corollaries/definitions without too many explanations: several terms that may only be familiar to experts in this line of works are not introduced with sufficient detail and many choices are not motivated.
>
> **A1**: We have revised our paper according to your valuable advice (please check the updated pdf), and we are happy to incorporate further explanations if needed.
>
> **Q2**: It is not clear why the assumption that trajectory returns are in [0,1] is reasonable in practice.
>
> **A2**: This assumption is consistent with prior studies on horizon-free RL, as it normalizes the return scale to eliminate artificial dependence on the horizon $H$. If trajectory returns are in $[0, R]$, we can transform the problem into an equivalent RL instance with returns in $[0, 1]$ by scaling all rewards $r$ to $r/R$. This transformation preserves the structure of the problem and does not alter its essence (e.g., the optimal policy).
>
> **Q3**. For offline RL, it is not clear why the assumption that the dataset contains trajectories is needed (lines 181-191). Can't you have only transitions?
>
>
> **A3**: We respectfully think there may be some misunderstandings. Our setup inherently accommodates the case where only transitions are available, as a full trajectory can always be decomposed into a collection of individual transitions. Furthermore, our algorithm (Algorithm 2) relies solely on transition-level information from the offline dataset, meaning that access to entire trajectories is not strictly necessary. We have added some discussions on this in the revised paper. Thanks for your question.
>
> **Q4**. The definition of Eluder dimension is not clear (eg what's an epsilon-independent sequence?). It is clearer in prior works
>
> **A4**: We acknowledge that prior works often define the
> $\epsilon$-independent sequence explicitly and seperately. In our work, this is integrated into the unified definition of the Eluder dimension, specifically in the last sentence: "i.e.,  there exists $g \in \Psi$ such that for all $t\in[L]$, $\sum_{l=1}^{t-1} |g(x^l)|^p \leq \epsilon^p$ and $|g(x^t)|>\epsilon$. " This reflects the properties of an
> $\epsilon$-independent sequence. We have revised the paper to make it clearer following your valuable advice.
>
> **Q5**. Why the bounds use the function class $\Psi$ defined at lines 258-259. Eg, why not using P itself, which after all is the hypothesis space used for learning?
>
> **A5**: First, in the definition of Eluder dimension (Definition 1), functions in $\Psi$ need to be mapped to scalars. We cannot just use $P$ as it is a mapping to a distribution, not a scalar. Second, we would like to clarify that the function class $\Psi$ is fundamentally based on $\mathcal{P}$ itself, as $P^*$ represents a fixed transition. Specifically, the function class $\Psi = {(s,a) \mapsto \mathbb{H}^2(P^*(s,a) ||P(s,a)) : P \in \mathcal{P}}$ is introduced to analyze the generalization from training to testing for the squared Hellinger distance between the ground-truth and estimated transitions. This construction is essential for deriving the bounds in our results. Such constructions are common in RL with function approximation. For instance, Wang et al. (2023) define $\Psi_h = {(x,a) \mapsto D_\triangle(f(x,a) || \mathcal{T}^{*,D}f(x,a)) : f \in \mathcal{F}}$ instead of directly using $\mathcal{F}$. Similarly, our use of $\Psi$ serves as an analytical tool for bounding generalization.
>
> **Q6**: In Section 4, it is never explicitly stated that the results before the paragraph "Extension to infinite class P" focus on finite P (the reader has to figure it out from the definition of beta)
>
> **A6**: Thanks for this comment, we have added explicit statements in the revised version.

---

> ### Author Response · Authors · 2024-11-21
>
> **Q7**: Not clear why using bracketing numbers in the analysis. Eg how do they compare to the more standard covering numbers (if there is any difference)? Also, it be great to give examples of what those are for "common" function classes (eg when P is linear)
>
> **A7**: Using bracketing numbers to deal with infinite function classes in MLE is quite standard, see [1], and prior RL theory work such as Wang et al. (2023), Wang et al. (2024), Liu et al. (2023),  Uehara \& Sun (2021). Unlike covering numbers, which count the minimum number of balls required to cover a function class, bracketing numbers are more suited for controlling the deviation between functions in the class by bounding the maximum distance between pairs of functions in a bracket, thereby providing finer control over the approximation in statistical learning theory. According to Appendix C.2 of Uehara \& Sun (2021), the bracketing number of Linear Mixture MDPs is $\mathcal{N}_{[]}(\epsilon,\mathcal{P},||\cdot||)=O(1/\epsilon)^d$. We have added this example following your advice.
>
> Reference
>
> [1] Empirical Processes in M-estimation. Vol. 6. Cambridge university press, 2000.
>
> **Q8**: Not clear how the "single-policy coverage" of Def. 3 compares to the more standard concentrability coefficients defined as ratios between policy visitation distributions used in the offline RL literature.
>
> **A8**: Our coverage definition shares similar spirits as the one in Ye et al. (2024). It is different from the definition of the ratio between policy visitation distributions. Both definitions reflect how well the state-action samples in the offline dataset $\mathcal D$ cover the state-action pairs induced by the comparator policy $\pi^\star$, but in different ways. It is hard to make direct comparisons. Note that the definition of the ratio between policy visitation distributions does not apply in our setting since $\mathcal D$ is not necessarily generated by some underlying fixed behavior policy. We also have some discussions of our coverage definition in Lines 461-469. We leave the derivation of horizon-free results with the coverage definition you mentioned as a future work.
>
> **Q9**: The algorithms studied here are both intractable due to the fact that they have to build a version space of transition kernels around the MLE. This is only tractable for simple function classes (eg finite) and limits the practical relevance of this work. Also, I feel this breaks a bit the overall story about the "simplicity" of the approaches. Eg, the conclusions state "we presented a minimalist approach for achieving nearly horizon-free and second-order bounds in online and offline RL: simply train transition models via Maximum Likelihood Estimation followed by optimistic or pessimistic planning", but I don't think this is properly correct: it gives the impression that all we have to do is MLE, but we do have to build version spaces around it, and that's not an easy thing.
>
> **A9**: We study the statistical efficiency of RL, the "simplicity" lies in the algorithm design and theoretical analysis, rather than the computational efficiency. Previous works on horizon-free or second-order RL need sophisticated algorithm design and analysis, while our algorithms and analysis are pretty simple and easy to understand. Our algorithms are standard and follow previous works (Algo.1 follows Liu et al. (2023),  Algo.2 follows Uehara \& Sun (2021)). We have added more clarifications on this point following your valuable advice.
>
>
> **Q10**: Lines 171-172: "Without loss of generality, we assume $V \in [0,1]$". I don't think this is without loss of generality, as the filtering approach proposed in footnote 2 would require doing policy-evaluation for many policies and transition kernels, which is what we want to avoid during training in the first place with an MLE approach
>
> **A10**: We respectfully think there may be some misunderstandings. We can do the filtering step in Line 6 of Algo.1. Specifically, when we do $(\pi^k, \widehat{P}^k) \leftarrow \arg\max_{\pi \in \Pi, P \in \widehat{\mathcal{P}}^k} V_{0; P}^\pi(s_0)$, we can just exclude those $V_{0; P}^\pi(s_0)$ that are not in $[0,1]$. This exclusion does not introduce additional computational costs because policy evaluations are inherently required in this step.
>
> We appreciate your comment about computational efficiency. We would like to emphasize that our primary focus remains on the statistical efficiency of standard model-based RL algorithms with function approximation, aligning with established theoretical frameworks extensively explored in the literature. Designing RL algorithms that are both computationally and statistically efficient with rich function approximation (even without horizon-free and second-order results) remains a long-standing open challenge, which is beyond the scope of this work.

---

> ### Author Response · Authors · 2024-11-21
>
> **Q11**: [minor] the abstract is very verbose, and I felt many details could have been omitted without altering its quality
>
> **A11**: We have shortened the abstract and removed extraneous details to improve readability in the updated version following your advice.
>
> **Q12**: The bound in Example 1 for tabular MDPs seems to have a worse dependence in SA that existing bounds for the same setting (see eg Jin et al. 2020 or Appendix C in Al-Marjani et al. 2023) despite having a better dependence in H. Any intuition why this happens?
>
> **A12**: These works only consider the tabular MDPs, while ours consider RL with general function approximations, and tabular MDP is just a special case in our setting. Some previous works on horizon-free RL also have much worse dependence in SA, for example, in Zhang et al. (2022), the regret depends on $\sqrt{S^9 A^3 K}$. We leave proving tighter bounds in SA when reducing to the special tabular setting as a future work.
>
> Again, we sincerely thank you for your time and constructive feedback. We hope our responses address your concerns, and we welcome any further questions or suggestions.

---

> > ### Comment · Reviewer_XvPg · 2024-11-26
> >
> > Thanks for the detailed response and for revising the paper in such a short time! The main reason behind my "borderline reject" score was the poor presentation, and I think the revised paper is clearer in many aspects (it does remain highly technical and quite hard to "digest", but I guess this is intrinsic in the topic). I have thus increased my score to 6. I don't consider increasing further as I still think that the intractability of the proposed algorithms is a major limitation.

---

> > > ### Author Response · Authors · 2024-11-27
> > > **Thanks for your positive feedback**
> > >
> > > Thanks for your positive feedback. We are glad that our responses addressed your concerns. Designing RL algorithms that are both computationally and statistically efficient with rich function approximation (even without horizon-free and second-order results) remains a long-standing open challenge, which we leave for future work. Again, thanks very much for your thoughtful review and constructive suggestions.
> > >
> > > Best regards,
> > >
> > > The Authors of Submission 11438

---

### Official Review · Reviewer_6vxv · 2024-11-04

**Soundness:** 3
**Presentation:** 3
**Contribution:** 3
**Rating:** 8
**Confidence:** 4

**Summary:**

This paper identifies minimalist reinforcement learning (RL) algorithms that can achieve nearly horizon-free and instance-dependent regret and sample complexity bounds,in both online and offline RL with non-linear function approximation. Based on a leverage of triangular discrimination, it demonstrates that existing standard model-based RL algorithms, using maximum likelihood estimation (MLE) for model learning and optimistic or pessimistic planning depending on the setting (online or offline), are sufficient to achieve these performance goals.

**Strengths:**

This paper explores horizon-free reinforcement learning (RL) within the context of general function approximations, which is a more challenging and generalized setting compared to prior work on tabular or linear MDPs. Theoretical analysis is solid, leveraging triangular discrimination to demonstrate that existing standard model-based algorithms can already achieve horizon-free regret. This analysis could be inspiring for future research in horizon-free RL. Furthermore, the paper provides a second-order instance-dependent upper bound, which can be particularly tight when the variance is low, yielding improved sample complexity in deterministic environments. Finally, the writing is logical, with ideas presented smoothly.

**Weaknesses:**

It's clear that this paper is purely focused on theoretical contributions. I think the primary weakness is the lack of intuitive insight into how triangular discrimination specifically assists in achieving horizon-free and second-order upper bounds. Although a proof sketch is provided, the underlying intuition remains somewhat unclear. A discussion on this would be appreciated.

**Questions:**

1. Are the assumptions listed on page 4 and in Definition 3 the only ones required for Theorem 2? I’m curious about this, as I would like to gain a better understanding of the factors contributing to the improved results compared to [1].

2. What do you think is the key barrier when attempting to apply a similar analysis to model-free methods? This question is not necessary to be replied.

[1] Uehara, M., & Sun, W. (2021). Pessimistic model-based offline reinforcement learning under partial coverage. arXiv preprint arXiv:2107.06226.

---

> ### Author Response · Authors · 2024-11-21
>
> We sincerely thank you for your positive feedback and thoughtful suggestions. Below, we provide detailed responses to your comments.
>
> **Q1**: I think the primary weakness is the lack of intuitive insight into how triangular discrimination specifically assists in achieving horizon-free and second-order upper bounds. Although a proof sketch is provided, the underlying intuition remains somewhat unclear. A discussion on this would be appreciated.
>
> **A1**: We would like to provide an intuitive explanation of how triangular discrimination specifically assists in achieving horizon-free and second-order upper bounds.
>
> The key tool we use is  Lemma 1, which upper bounds the difference between the means of two distributions by a term related to the variance of one of the distributions and the triangular discrimination. After applying the simulation lemma, this result allows us to bound the difference of $V$-values by the variance of $V$ and the triangular discrimination between $V$-values. This is where the second-order bound arises.
>
> Furthermore, using $ D_\triangle \leq 4 \mathbb{H}^2$ (where $\mathbb{H}^2$ denotes the squared Hellinger distance) and the information processing inequality, we can bound the triangular discrimination between $V$-values by the squared Hellinger distance between the ground-truth transition $P$ and the estimated transitions $\hat P^k$. The sum of Hellinger distance terms is then upper bounded using the MLE generalization bound to achieve the horizon-free result.
>
> The key steps leveraging triangular discrimination can be found in Lines 376-381 of the paper (updated version). We hope this explanation clarifies the role of triangular discrimination in our analysis.
>
>
> **Q2**: Are the assumptions listed on page 4 and in Definition 3 the only ones required for Theorem 2? I’m curious about this, as I would like to gain a better understanding of the factors contributing to the improved results compared to [1].
>
> **A2**: Yes, the assumptions listed on page 4 and in Definition 3 are the only ones required for Theorem 2.
>
> It is worth noting that Definition 3 differs from the coverage definition in [1]. Specifically, the denominator in the coverage definition from [1] is given by $\mathbb{E}_{s,a \sim d^{\pi^b}_h} \mathbb{H}^2\left( P(s,a) || P^\star(s,a) \right) $, where $\pi^b$ is the fixed behavior policy used to collect $\mathcal{D}$. This definition is not applicable to our setting, as our dataset $\mathcal{D}$ is not necessarily generated by a fixed behavior policy.
>
> On the other hand, our horizon-free result does not hold in the setting of [1], where $\mathcal{D}$ is collected using a fixed behavior policy $\pi^b$ and the concentrability coefficient is defined in their specific way. Extending our horizon-free results to the setting in [1] remains an open problem and a potential direction for future work.
>
>
> **Q3**: What do you think is the key barrier when attempting to apply a similar analysis to model-free methods? This question is not necessary to be replied.
>
> **A3**: Although this question is optional, we would like to share our thoughts, as this is an important area for further exploration.
>
> In model-free methods, the key challenge lies in the absence of an explicit model of transition dynamics. Without an estimated model, the simulation lemma and associated variance conversion lemmas (Lemma 13 and Lemma 16) used in our work are not directly applicable.
>
> Specifically:
>
> In model-free settings, recent work (e.g., Wang et al. (2024)) has achieved second-order bounds under a non-conventional distributional RL framework. However, their approach introduces additional polynomial factors of
> H during variance transformations. To be more precise, they will introduce poly(H) factors when trying to upper bound the intermediate term $\mathbb{E}_{\pi,x_1}[Var(f_h(x_h,a_h))]$ by $Var(Z^\pi(x_1))$ (please see Section 5.2 of Wang et al. (2024) for details). Therefore, it is unclear how to get the horizon-free bounds in their approach.
>
>
>  In contrast, in our model-based approach, we only need to upper bound $\big(\mathbb V_{P^\star} V_{h+1; \widehat P^k}^{\pi^k}\big)(s_h^k, a_h^k)$ by $\big(\mathbb V_{P^\star} V_{h+1}^{\pi^k}\big)(s_h^k, a_h^k)$ to get the desired second-order bound, which can be done without introducing additional H factors by leveraging Lemma 13.
>
> Designing model-free methods capable of achieving horizon-free and second-order bounds remains a challenging and exciting direction for future research. We believe our work provides valuable insights that could inform such developments, and we look forward to exploring this area further.
>
> Thank you again for your positive, valuable feedback and thought-provoking questions. We hope our responses provide clarity and address your concerns.

---

> > ### Comment · Reviewer_6vxv · 2024-11-22
> >
> > I thank the authors for their thoughtful response. I have read the revision and authors' replies to all reviewers, and will increase my score to 8.

---

> > > ### Author Response · Authors · 2024-11-23
> > >
> > > Thank you for your positive feedback!
> > >
> > > Best regards,
> > > The Authors of Submission 11438

---

### Official Review · Reviewer_dSXD · 2024-11-05

**Soundness:** 3
**Presentation:** 3
**Contribution:** 3
**Rating:** 8
**Confidence:** 4

**Summary:**

This paper presents a minimalist model-based RL approach focused on achieving horizon-free and second-order bounds in both online and offline RL contexts. The authors advocate for using MLE for transition model learning, combined with optimistic planning in online RL and pessimistic planning in offline RL. The primary theoretical contribution is demonstrating that this straightforward MLE-based approach, which doesn’t require complex, variance-weighted learning procedures, achieves near-horizon-free sample complexity and second-order regret bounds. The analysis shows that this approach is effective in handling scenarios with either non-linear function approximations or time-homogeneous, low-variance policies, which benefit from the instance-dependent nature of second-order bounds.

**Strengths:**

1. The theoretical results show that performance does not heavily depend on the planning horizon, making this approach comparable to contextual bandits in complexity. The second-order bounds, scaling with policy variance, are particularly advantageous for nearly deterministic systems or policies with low variance.

2. The approach is designed to be versatile, working effectively in online, offline, and hybrid settings.

**Weaknesses:**

1. While the approach is theoretically sound, its reliance on computationally intensive MLE optimization could hinder scalability for larger or more complex model classes. To address this, it may be valuable to explore specific approximate MLE methods, such as variational inference or stochastic optimization. Alternatively, a discussion of computational complexity trade-offs could provide further clarity and practical guidance.

2. The paper presents strong theoretical contributions, but the lack of empirical validation on challenging RL tasks limits its practical impact. It would be helpful to evaluate the approach to environments such as tabular MDPs. Including such experiments would significantly strengthen the paper’s practical contributions.

**Questions:**

1. Given that the current approach is theoretically sound but may be computationally intense, are there approximate methods that could retain accuracy while reducing computational demands?

2. What are the implications if the assumed model class does not include the true dynamics of the environment? Could the algorithm's performance degrade, and if so, how might this be mitigated?

---

> ### Author Response · Authors · 2024-11-21
>
> We sincerely appreciate your positive feedback and valuable suggestions. Below, we address your comments in detail.
>
>
> **Q1**: While the approach is theoretically sound, its reliance on computationally intensive MLE optimization could hinder scalability for larger or more complex model classes. To address this, it may be valuable to explore specific approximate MLE methods, such as variational inference or stochastic optimization. Alternatively, a discussion of computational complexity trade-offs could provide further clarity and practical guidance.
>
> **A1**: Thank you for your thoughtful and constructive feedback. We appreciate your concern regarding the computational intensity of MLE optimization and the suggestion of discussing approximate MLE methods.
>
> **Addressing Computational Scalability:**
>
> While our current focus is on the statistical efficiency of RL with function approximation, we acknowledge the importance of improving computational efficiency. To address the computational demands of MLE, we propose exploring approximate MLE methods, particularly leveraging recent advancements in Diffusion Models:
>
> (1) Theoretical Support for Diffusion Models:
> In [1], it is demonstrated that Diffusion Models can serve as effective approximations to MLE by providing an upper bound on the squared Hellinger distance between the estimated distribution and the ground-truth distribution. This theoretical foundation suggests that Diffusion Models can reliably approximate the ground-truth transition, which is crucial for our proofs that require bounding the squared Hellinger distance.
>
> (2) Empirical Evidence in RL Contexts:
> Empirical studies, such as [2], have successfully integrated Diffusion Models for approximate MLE in RL settings. These works highlight the practical potential of Diffusion Models to enhance RL algorithms' scalability and efficiency.
>
> **Future Directions:**
>
> Incorporating Diffusion Models as an approximate MLE method represents a promising direction to balance computational and statistical efficiency. However, developing and validating such integrations within our framework constitutes a substantial extension of our current work. As a result, we have identified this as an exciting direction for future research.
>
> **Scope of Current Work:**
>
> Our primary focus remains on the statistical efficiency of standard model-based RL algorithms with function approximation, aligning with established theoretical frameworks extensively explored in the literature. Designing RL algorithms that are both computationally and statistically efficient with rich function approximation (even without horizon-free and second-order results) remains a long-standing open challenge, which is beyond the scope of this work. Addressing computational complexity trade-offs and integrating approximate MLE methods like Diffusion Models, while valuable, extend beyond our current objectives.
>
> We appreciate your insightful suggestions and believe that exploring these approaches will significantly advance scalable and efficient RL methodologies.
>
> References:
>
> [1] Sampling is as easy as learning the score: theory for diffusion models with minimal data assumptions. ICLR 2023.
>
> [2] Diffusion Spectral Representation for Reinforcement Learning. NeurIPS 2024.
>
>
> **Q2**: The paper presents strong theoretical contributions, but the lack of empirical validation on challenging RL tasks limits its practical impact. It would be helpful to evaluate the approach to environments such as tabular MDPs. Including such experiments would significantly strengthen the paper’s practical contributions.
>
> **A2**: We appreciate the feedback regarding empirical validations. Our primary focus in this work is on statistical efficiency, and we believe that our contribution on the theoretical side is already novel and significant. We will consider studying experimental validation in the future (possibly together with the diffusion model investigation mentioned above).
>
> We appreciate your suggestion and are committed to exploring these empirical avenues to complement our theoretical contributions in subsequent research.

---

> ### Author Response · Authors · 2024-11-21
>
> **Q3**: Given that the current approach is theoretically sound but may be computationally intense, are there approximate methods that could retain accuracy while reducing computational demands?
>
> **A3**: Thanks very much for this valuable question.
>
> First, as mentioned above, we believe incorporating diffusion models as an approximate MLE method represents a promising direction for balancing computational and statistical efficiency (please refer to **A1** for more discussions). We leave this as an important future work.
>
> Second, we want to emphasize again that our primary focus remains on the statistical efficiency of standard model-based RL algorithms with function approximation, aligning with established theoretical frameworks extensively explored in the literature. Designing RL algorithms that are both computationally and statistically efficient with rich function approximation (even without horizon-free and second-order results) remains a long-standing open challenge, which is beyond the scope of this work.
>
>
> Thanks for your insightful comment, we will add more discussions regarding this.
>
>
>
> **Q4**: What are the implications if the assumed model class does not include the true dynamics of the environment? Could the algorithm's performance degrade, and if so, how might this be mitigated?
>
> **A4**: Thank you for this valuable question. Below, we discuss the implications of model misspecification and potential mitigation strategies:
>
> **1. Impact of Model Misspecification:**
>     When the model class does not include the true dynamics, the original algorithms may fail to achieve the desired bounds. Model misspecification introduces challenges, as the optimism/pessimism guarantees derived from the version spaces may no longer hold. This can degrade the algorithm's performance in terms of regret or sample complexity.
>
> **2. Mitigation Strategies:**
>     To handle model misspecification, the algorithms can be modified to incorporate a relaxation that accounts for the misspecification level. For ease of interpretation, we take offline RL as an example. We need to adjust the threshold parameter $\beta$ in Line 2 of Algorithm 2 based on the misspecification level to get an approximate pessimism guarantee. Specifically, following Foster et al. (2024), we define the misspecification level as
> $
> \epsilon_*:=\max_{s,a}\min_{P\in\mathcal{P}}D_{\chi^2}(P^*(s,a)||P(s,a))
> $.
>  Then, we need to set $\beta=4\log(|\mathcal{P}|/\delta)+KH\epsilon_*$ to enable Algo.2 to tolerate some degree of misspecification. The resulting performance gap is given by:
> \begin{align}
>     V^{\pi^*}-V^{\widehat\pi} \leq O\Big(\sqrt{C^{\pi^*} Var_{\pi^*}(\log(|\mathcal{P}|/\delta)+H\log(1+\epsilon_*))/{K} } + C^{\pi^*}(\log(|\mathcal{P}|/\delta)+H\log(1+\epsilon_*))/{K}+H\epsilon_*+H\sqrt{\epsilon_*}\Big).
> \end{align}
>
> The suboptimality gap will have some additional error terms related to the level of model misspecifications, which is expected and characterizes the effects of model misspecifications. We conjecture that the multiplicative H terms on $\epsilon_*$ are unavoidable, and one potential way to mitigate it is to define the misspecification over whole trajectory (i.e., divergence between the whole trajectory distributions), which we leave for future work.
>
> We thank you again for this insightful question and look forward to incorporating these ideas into future research.
>
> Again, thank you very much for reviewing our work and your positive comments. We are happy to provide further clarifications if needed.

---

> > ### Comment · Reviewer_dSXD · 2024-11-26
> >
> > I thank the authors for addressing my questions. I have decided to increase the score to 8.

---

> ### Author Response · Authors · 2024-11-27
> **Thanks for your positive feedback**
>
> Thank you for your positive feedback. We are glad that our responses addressed your concerns. Again, thanks very much for your thoughtful review and constructive suggestions.
>
> Best regards,
>
> The Authors of Submission 11438

---

### Official Review · Reviewer_ttab · 2024-11-05

**Soundness:** 3
**Presentation:** 2
**Contribution:** 3
**Rating:** 6
**Confidence:** 4

**Summary:**

This paper proposes a minimalist approach to Model-Based Reinforcement Learning (RL) that leverages Maximum Likelihood Estimation (MLE) for learning transition models and integrates optimistic and pessimistic planning. The proposed scheme achieves strong regret and sample complexity bounds in both online and offline settings. It also provides nearly horizon-free and second-order bounds under certain conditions.

**Strengths:**

1. The authors provide solid theoretical analysis.

2. The methodology stands out due to its simplicity. The authors focus on using standard techniques (MLE, version space, optimistic/pessimistic planning) without introducing complex or specialized methods.

**Weaknesses:**

1. While the paper mentions the simplicity and standard nature of the algorithm, it lacks a direct comparison with other recent state-of-the-art Model-Based RL methods. A more explicit comparison would help highlight the contributions and potential advantages of this approach.

2. While the paper emphasizes it provides a nearly horizon-free result, I do not think it is a novel contribution with the assumption $r(s_h, a_h) \in [0,1 / H]$. I am more interested in approaches of [1,2], which also remove $\log T$ in their regret bound. Can you compare your work with these two papers?

3. The theoretical conclusions of this article are too dense and lack discussion.

References:

[1] Zhang, Zihan, Xiangyang Ji, and Simon Du. "Horizon-free reinforcement learning in polynomial time: the power of stationary policies." Conference on Learning Theory. PMLR, 2022.

[2] Li, Shengshi, and Lin Yang. "Horizon-free learning for Markov decision processes and games: stochastically bounded rewards and improved bounds." International Conference on Machine Learning. PMLR, 2023.

**Questions:**

1. As I have mentioned in (2) in weaknesses, can you compare your horizon-free results with [1,2]?

2. The paper discusses the use of MLE for transition modeling. How does this choice impact the trade-offs between computational efficiency and the accuracy of the learned model in comparison to other model learning techniques (e.g., Bayesian approaches)?

3. How does the use of MLE for model learning affect the sensitivity of the model to noise in the environment, particularly in settings where the transitions are stochastic or partially observable? Can the methodology be adapted to handle such cases without compromising the theoretical bounds?

---

> ### Author Response · Authors · 2024-11-21
>
> We sincerely thank you for your valuable feedback and suggestions. Our responses are as follows, which we hope could well address your concerns.
>
> **Q1**: While the paper mentions the simplicity and standard nature of the algorithm, it lacks a direct comparison with other recent state-of-the-art Model-Based RL methods. A more explicit comparison would help highlight the contributions and potential advantages of this approach.
>
> **A1**: We thank the reviewer for highlighting the importance of comparisons. To the best of our knowledge, the most relevant recent works on horizon-free bounds in Model-Based RL are Zhao et al. (2023b) and Huang et al. (2024), both of which we cite in our related work.
>
> Zhao et al. (2023b) focuses on linear mixture MDPs in the online setting and achieves horizon-free and second-order bounds. However, our work targets RL with general function approximation and addresses both online and offline settings, making it broader in scope.
>
> Huang et al. (2024) achieves results similar to ours in the online setting with function approximation but employs complex techniques (e.g., variance-weighted regression). In contrast, our work relies on a simpler MLE-based approach while achieving comparable results. Notably, we extend our analysis to the offline setting, which neither Zhao et al. (2023b) nor Huang et al. (2024) address.
>
> We will incorporate these comparisons and emphasize our work's simplicity and offline contributions in the final version.
>
> **Q2**: While the paper emphasizes it provides a nearly horizon-free result, I do not think it is a novel contribution with the assumption $r(s_h,a_h)\in[0,1/H]$.
>
> **A2**: We believe there may be some misunderstandings. We **only assume $r(\tau) \in [0,1]$** for any trajectory $\tau:= \{s_0,a_0,\dots, s_{H-1},a_{H-1}\}$ where $r(\tau)$ is short for $\sum_{h=0}^{H-1} r(s_h,a_h)$. Note that this setting is more general than assuming each one-step reward is bounded, i.e., $r(s_h,a_h) \in [0,1/H]$, and allows to represent the sparse reward setting. Notably, this setting aligns with [1].
>
>
> **Q3**: Can you compare your horizon-free results with [1,2]?
>
> **A3**: Thank you for the insightful suggestion to compare our work with [1] and [2]. While both [1] and [2] focus on the tabular online RL setting, our work addresses RL with function approximation, encompassing both online and offline RL. Additionally, our results feature second-order bounds, further distinguishing the scope and novelty of our contributions.
>
> The results in [1] demonstrate that the logarithmic dependence on the planning horizon $H$ can be removed in the tabular setting. However, this improvement introduces additional polynomial dependence on $S$ (number of states) and $A$ (number of actions). Specifically, the regret bound in [1] scales as $O(\sqrt{S^9 A^3 K})$, where $K$ is the number of episodes. By contrast, when applied to the tabular setting, our results yield a regret bound of $O(\sqrt{S^3 A^2 \sum_k \mathrm{Var}_{\pi^k}} \log H)$, which has better dependence on $S$ and $A$, while also incorporating second-order terms.
>
> The follow-up work [2] improves upon [1], achieving a regret bound of $O(S \sqrt{A K})$ in tabular online MDPs, also eliminating the logarithmic dependence on $H$. Moreover, [2] relaxes the assumption on bounded rewards by allowing the expected total reward to satisfy: $|r(s_h, a)| \leq 1$, and $E_{\pi} \left[ \sum_{t=h}^H |r(s_h, a_h)| \right] \leq 1$, for all $h \in [H]$, $h$-reachable states $s_h$, and policies $\pi$. While our work does not consider this relaxed reward setting, extending our results to accommodate such scenarios represents an interesting direction for future research. Nonetheless, it is important to emphasize that the results in [2] do not provide second-order regret bounds and are limited to online tabular MDPs, whereas our work addresses both online and offline RL with function approximation.
>
> In summary, while [1] and [2] make significant contributions to the tabular online RL setting, our work is distinct in its focus on RL with function approximation, addressing both online and offline RL. This broader scope, along with the incorporation of second-order bounds, sets our contributions apart. We will add further discussions on [1] and [2] following your advice.
>
> **Q4**: Dense theoretical conclusions and lack of discussion.
>
> **A4**: Thanks very much for the advice. We will add more discussions of our theoretical conclusions. If there are specific aspects you believe require further elaboration, we would be grateful for your guidance.

---

> > ### Comment · Reviewer_ttab · 2024-11-21
> >
> > I thank the author for the detailed response, which addresses my questions. I consider raising my score to 6.

---

> > > ### Author Response · Authors · 2024-11-22
> > > **Thanks for your positive feedback**
> > >
> > > Dear Reviewer ttab,
> > >
> > > Thank you very much for your positive feedback and for taking the time to thoroughly review our work. We are delighted to hear that our response has satisfactorily addressed your concerns. We deeply appreciate your consideration of raising your score, and we value your insights, which have contributed to improving the clarity and impact of our submission.
> > >
> > > Best regards,
> > >
> > > The Authors of Submission 11438

---

> ### Author Response · Authors · 2024-11-21
>
> **Q5**: The paper discusses the use of MLE for transition modeling. How does this choice impact the trade-offs between computational efficiency and the accuracy of the learned model in comparison to other model learning techniques (e.g., Bayesian approaches)?
>
> **A5**: While the MLE approaches in this work are statistically efficient, as noted in the Conclusion section, our algorithms are not computationally tractable due to the need for optimism/pessimism planning. Designing computationally efficient RL algorithms with rich function approximation (even without horizon-free and second-order results) remains a long-standing open challenge, which is beyond the scope of this work.
>
> We are not entirely certain what you mean by "Bayesian approaches," as this term can cover a wide range of techniques. If you are referring to the posterior sampling framework proposed by Osband et al. (2013), such methods typically require access to the model distribution as a prior. In contrast, our approach does not rely on such a prior, which broadens its applicability, particularly in settings where defining or accessing a prior is impractical. In terms of trade-offs, Bayesian approaches like posterior sampling often involve maintaining and sampling from a posterior distribution, which can be computationally expensive, especially in high-dimensional or complex environments. Our method avoids this computational overhead, offering better efficiency.
>
> Reference
>
> Osband, I., Russo, D., \& Van Roy, B. (2013). (More) efficient reinforcement learning via posterior sampling. Advances in Neural Information Processing Systems, 26.
>
> **Q6**: How does the use of MLE for model learning affect the sensitivity of the model to noise in the environment, particularly in settings where the transitions are stochastic or partially observable? Can the methodology be adapted to handle such cases without compromising the theoretical bounds?
>
> **A6**: First, our paper **does focus on the stochastic transitions**.
> For example, in the online setting, in Theorem 1 works for the stochastic setting.
> Note that when the transitions becomes more deterministic, the regret becomes even better.
>
> Second, partial observability is out of the scope of this paper. But for POMDPs, Liu et al. (2023) demonstrated that OMLE (similar to Algorithm 1) can learn near-optimal policies with polynomial sample complexity. Extending our horizon-free and second-order theoretical results to POMDPs is a promising future direction, which we plan to pursue.
>
> Again, we sincerely thank you for your time and constructive feedback. We hope our responses address your concerns, and we welcome any further questions or suggestions.

---

### Meta-Review · Area_Chair_17J1 · 2024-12-20

**Metareview:**

Model-based RL as a Minimalist Approach to Horizon-Free and Second-Order Bounds

Summary: The paper introduces a minimalist approach to model-based reinforcement learning (MBRL) that combines Maximum Likelihood Estimation (MLE) for transition modeling with optimistic and pessimistic planning, tailored for online and offline RL settings. It emphasizes achieving nearly horizon-free and second-order regret bounds, offering statistical efficiency akin to contextual bandits. This is achieved without the need for new algorithms but by leveraging triangular discrimination techniques. The work also explores performance improvements under deterministic transitions and proposes bounds for finite and infinite model classes using Eluder dimension and bracketing numbers.

Comments: We received 4 expert reviews, with the scores 6, 6, 8, 8, and the average score is 7.0. The reviewers are positive about the technical contributions of this paper. They appreciated the simplicity of the approach, of leveraging existing model-based RL methods, using Maximum Likelihood Estimation (MLE) and standard planning techniques, to achieve strong theoretical guarantees. I also believe that this is a valuable result for the RL community. The technical contribution in establishing a nearly horizon-free and variance-sensitive regret bound is also original and important. The paper is also well-written.

The reviewers have also provided multiple comments to strengthen the paper. This includes more comparisons with some specific related works and providing more explanation of the true practicality of implementing the MLE-based approach (line 5 in the algorithm). I recommend the authors update the paper by accommodating these suggestions.

**Additional Comments On Reviewer Discussion:**

Please see the "Comments" in the meta-review.

---

### Decision · Program_Chairs · 2025-01-22

Accept (Poster)